# Using avalanche problems to examine the effect of large-scale atmosphere-ocean oscillations on avalanche hazard in western Canada

Pascal Haegeli[1], Bret Shandro[1,2], Patrick Mair[3]

[1]School for Resource and Environmental Management, Simon Fraser University, Burnaby, V5T 2P9, Canada
[2]6 Point Engineering and Avalanche Consulting, Nelson, V1L 4H5, Canada
[3]Dept. Psychology, Harvard University, Cambridge MA, 02138, United States

*Correspondence to*: Pascal Haegeli (pascal_haegeli@sfu.ca)

**Abstract.** Numerous large-scale atmosphere-ocean oscillations including the El Niño-Southern Oscillation (ENSO), the Pacific Decadal Oscillation (PDO), the Pacific North American Teleconnection Pattern (PNA) and the Artic Oscillation (AO) are known to substantially affect winter weather patterns in western Canada. Several studies have examined the effect of these oscillations on avalanche hazard using long-term avalanche activity records from highway avalanche safety programs. We present a new approach for gaining additional insight into these relationships that uses avalanche problem information published in public avalanche bulletins during the winters of 2010 to 2019. For each avalanche problem type, we calculate seasonal prevalence values for each forecast area, elevation band and season, which are then included in a series of beta mixed-effects regression models to explore both the overall and regional effects of the Pacific-centered oscillations (PO; including ENSO, PDO, PNA) and AO on the nature of avalanche hazard in the study area. We find significant negative effects of PO on the prevalence of *Storm slab avalanche problems*, *Wind slab avalanche problems*, and *Dry loose avalanche problems*, which agree reasonably well with the known impacts of PO on winter weather in western Canada. The analysis also reveals a positive relationship between AO and the prevalence of *Deep persistent slab avalanche problems* particularly in the Rocky Mountains. In addition, we find several smaller-scale patterns that highlight that the avalanche hazard response to these oscillations varies regionally. Even though our study period is short, our study shows that the forecaster judgment included in avalanche problem assessments can add considerable value for these types of analyses. Since the predictability of the most important atmosphere-ocean oscillations is continuously improving, a better understanding of their effect on avalanche hazard can contribute to the development of informative seasonal avalanche forecasts in a relatively simple way.

## 1 Introduction

Snow avalanches are an inherent natural hazard in mountainous regions that receive substantial amounts of seasonal snow. In these regions, snow avalanches can threaten communities, transportation corridors, critical infrastructure (e.g., hydroelectric dams, transmission and communication lines, pipelines) and resource extraction operations. In Western countries, most

people killed in avalanches are recreationists pursuing winter mountain activities such as backcountry skiing, mountain snowmobile riding and out-of-bounds skiing. Avalanche hazard conditions continuously evolve in response to the weather conditions experienced during a winter. Much of existing avalanche research is focused on examining the short-term effects of weather on avalanche conditions to support operational avalanche forecasting. However, examining the relationship between longer-term variations in weather patterns and the nature of avalanche hazard can also offer valuable insight that can lead to the development of seasonal avalanche hazard forecasts (McClung, 2013) and contribute to our understanding of the effect of climate change on avalanche hazard.

The winter weather in western Canada is affected by several prominent large-scale atmosphere-ocean oscillations including the El Niño-Southern Oscillation, the Pacific Decadal Oscillation, the Pacific North American Teleconnection Pattern, and the Arctic Oscillation. Since the effects of these large scale atmosphere-ocean oscillations on winter temperature and precipitation patterns in the region are well understood (e.g., Shabbar and Bonsal, 2004; Stahl et al., 2006; Fleming and Whitfield, 2010), it is no surprise that numerous studies have examined the effect of these weather patterns on the seasonal avalanche hazard conditions in the area. Fitzharris (1987) was the first in Canada to consider anomalies in atmospheric circulation patterns to explain major avalanche winters in Rogers Pass, BC. McClung (2013) found significant correlations between avalanche activity (overall, as well as dry-snow and wet-snow avalanches separately) with positive El Niño-Southern Oscillation phase winters at Bear Pass and Kootenay Pass, British Columbia. Most recently, Thumlert et al. (2014) confirmed these results in their study examining the correlation between large-scale climate oscillations and yearly avalanche activity at six highway programs in British Columbia (Bear Pass, Coquihalla, Duffy Lake, Kootenay Pass, Ningunsaw, and New Denver to Kaslo). In addition, they found a similarly significant relationship between avalanche activity and the Pacific Decadal Oscillation, with more wet avalanches during positive/warmer phase winters and more dry avalanches during negative/colder phase winters. Thumlert et al. (2014) also identified a positive correlation between the North Atlantic Oscillation, a climate oscillation related to the Arctic Oscillation (Bjerknes, 1964), and the frequency of wet slab avalanches. Similar studies have been conducted in other geographic regions including Iceland (Keylock, 2003) and the Pyrenes in Northern Spain (García-Sellés et al., 2010).

While the Canadian studies offer valuable insight into the effect of atmosphere-ocean oscillations on the nature of avalanche hazard in western Canada, they also have limitations. For example, since all these studies focused on avalanche observations from highway avalanche safety programs, they only represent point observations and are unable to provide a comprehensive perspective on the overall effect across western Canada. Furthermore, changes in avalanche risk mitigation practices along these transportation corridors can add noise to avalanche activity records that make it more difficult to attribute the observed patterns to changes in winter weather (Bellaire et al., 2016; Sinickas et al., 2016; Jamieson et al., 2017). Furthermore, the seasonal magnitude of avalanche activity, even if separated into dry and wet avalanches, only provides a limited perspective on the nature of avalanche hazard.

The objective of the present study is to complement the existing research on the effect of large-scale atmosphere-ocean oscillations on avalanche hazard in western Canada by taking advantage of the avalanche problem information included in

public avalanche bulletins that follow the conceptual model of avalanche hazard (Statham et al., 2018a). This approach links the analysis more closely to backcountry avalanche risk management and overcomes some of the shortcomings of previous studies. Even though linking avalanche hazard conditions to large-scale atmosphere ocean oscillations is unable to provide direct insight for operational, day-to-day avalanche safety decisions, a better understanding of these relationships has the potential to allow the avalanche safety community to take advantage of atmosphere-ocean oscillation predictions that are routinely provided by meteorological services to produce informative seasonal avalanche hazard forecasts. Being able to predict the general nature of seasonal avalanche conditions (e.g., there is a good chance that this winter will be dominated by a deep persistent avalanche problem) would help avalanche professionals and recreationists to develop meaningful risk management expectations for an upcoming season. As pointed out by LaChapelle (1980) and McClung (2002), avalanche forecasting is a dynamic and iterative process that resembles Bayesian updating where having a prior or hypothesis is critical.

## 2 Background

### 2.1 Atmosphere-ocean oscillations affecting winter weather in western Canada

The most prominent large-scale atmosphere-ocean oscillations affecting the winter weather in western Canada and the Pacific Northwest of the United States is the El Niño-Southern Oscillation (ENSO), which originates from an irregular fluctuation between unusually warm (El Niño) and unusually cold (La Niña) conditions in the Eastern South Pacific off the coast of Peru (McPhaden et al., 2006). El Niño and La Niña events typically occur every two to seven years and have large effects on the weather in numerous regions around the world. In western Canada and the Pacific Northwest, El Niño winters are associated with a shift towards warmer than normal temperatures, whereas La Niña winters are colder than normal (Shabbar and Khandekar, 1996; Shabbar and Bonsal, 2004; Bonsal et al., 2001). The signal in precipitation is less distinct. Shabbar et al. (1997) did not identify any precipitation anomalies during El Niño or La Niña winters in western Canada, but found negative anomalies for the winters following the onset of an El Niño, and positive anomalies following a La Niña event. Lute and Abatzoglou (2014) showed that La Niña events in the Pacific Northwest are associated with more frequent and more intense snowfall events. Numerous studies (e.g., Fleming and Whitfield, 2010; Wise, 2010; Jin et al., 2006) have shown that these general patterns in ENSO anomalies are blurred by considerable regional differences and temporal variabilities. Stahl et al. (2006), for example, showed that the coastal regions of British Columbia (BC) exhibit a stronger temperature response while BC's interior shows a stronger response in the precipitation patterns. Fleming and Whitfield (2010) highlight that the positive temperature signal of El Niño is weaker in northern BC, and while El Niño tends to bring drier conditions to the southern part of BC, it produces wetter conditions along the northern coast. McAfee and Wise (2016) suggest that the effects of ENSO are stronger in late winter than early winter.

The Pacific Decadal Oscillation (PDO) (Mantua and Hare, 2002; Newman et al., 2016), a primarily interdecadal atmosphere-ocean oscillation linked to changes in the sea surface temperatures in the northern mid-latitude Pacific basin, is primarily

known for its modulating effect of ENSO related temperature anomalies. The positive temperature anomalies during El Niño winters are stronger and more widespread during positive PDO winters (Mantua and Hare, 2002; Bonsal et al., 2001) and simultaneously occurring negative ENSO and PDO phases have been linked to negative temperature and increased precipitation anomalies in western Canada (Bonsal et al., 2001; Stahl et al., 2006; Fleming and Whitfield, 2010).

The Pacific North America Teleconnection Pattern (PNA) (Leathers et al., 1991) is a climate oscillation that affects temperature and precipitation distribution over the Pacific and North America by modulating the jet stream and storm tracks over the region on intraseasonal and interannual time scales. Relevant for western Canada, the positive pattern is generally associated with an anomalously deep Aleutian low and an enhanced ridge over western North America, which leads to a more meridional flow pattern with warmer and drier air and reduced snow cover. The negative PNA pattern has a more zonal

circulation pattern, colder than average temperatures and produce higher snow accumulation (Kluver and Leathers, 2015; Brown and Goodison, 1996; Stahl et al., 2006; Wallace and Gutzler, 1981).

Another atmosphere-ocean oscillation affecting the winter weather in western Canada and the Pacific Northwest is the Arctic Oscillation (AO; Thompson and Wallace, 1998), which is distinctly different from the Pacific orientated teleconnections mentioned previously (Wu and Hsieh, 2004). The AO is a hemispheric-scale climate oscillations that mostly affects higher

latitudes and represents differences in atmospheric mass between the Arctic and mid-latitudes on month-to-month timescales (Thompson and Wallace, 1998). Positive AO anomalies with lower pressure over the Arctic and higher pressure in mid latitudes result in stronger westerly flows and higher springtime temperatures in northwestern BC, while negative phase AO conditions have weaker meridional pressure gradients and therefore exhibit weaker westerly flows (Fleming et al., 2006; Moore et al., 2009). Gobena et al. (2013), who studied the effect of AO on stream flows in the Columbia River Basin of

Southeastern BC, only identified effects during negative AO anomalies with cooler than average temperatures during December, January and March, and below-average precipitation during winter and spring. Vincent et al. (2015), on the other hand, noted a positive association of winter temperatures in Northern BC with the North Atlantic Oscillation, a close relative to the AO (Fleming and Dahlke, 2014a). They did not find a significant signal in winter precipitation.

## 2.2 A meaningful characterization of avalanche winters

One of the challenges for examining the relationship between atmosphere-ocean oscillations and the seasonal avalanche hazard is how to describe avalanche hazard in a meaningful way. While existing studies have primarily focused on the frequency of avalanches, the ratio between dry and wet avalanches, or the number of avalanche cycles, Atkins (2004) and Statham et al. (2018a) highlighted that the nature of avalanche hazard, its distribution in the terrain and evolution throughout the season are much more important for avalanche risk management than the frequency of avalanches alone. The presence of

a persistent weak layer in the snowpack can dominate the nature of an avalanche winter even if the number of associated avalanches is relatively small (Haegeli and McClung, 2007). Avalanche professionals therefore commonly label winters according to their standout avalanche hazard characteristic (e.g., exceptional number of surface hoar layers, early November

facet-rain crust combination). Hence, examining the relationship between long-term atmosphere-ocean oscillations and avalanche hazard meaningfully requires a more comprehensive way to describe the nature of avalanche winters.

Avalanche hazard assessments included in public avalanche bulletins offer a more comprehensive perspective on avalanche hazard than avalanche observations alone. When preparing bulletins, human forecasters assimilate a wide range of observations and assessments to develop a detailed picture of the regional hazard conditions. This human contribution circumvents some of the challenges of pure avalanche observation datasets. For example, human forecasters know that there are direct action avalanches during a storm even if poor visibility prevents the observation of these avalanches. Human

forecasters can also make informed extrapolations over space and time. However, the qualitative nature of the hazard description in avalanche bulletins has traditionally prevented its use in systematic climate analyses.

Since the winter of 2010, public avalanche forecasters in Canada have been using the conceptual model of avalanche hazard (CMAH; Statham et al., 2018a) to document their assessments more systematically. The CMAH identifies key components of avalanche hazard and structures them in a systematic workflow to provide a meaningful pathway for synthesizing

available avalanche safety observations (weather, snowpack and avalanche observations), conceptualizing hazard conditions and choosing appropriate risk treatment actions. A key component of the CMAH is the identification and characterization of avalanche problems (Haegeli et al., 2010; Lazar et al., 2012), which represent operational avalanche safety concerns that emerge from the preceding weather and snowpack conditions. Avalanche hazard assessments typically include one or more avalanche problems, which are described in terms of their avalanche problem type, where they can be found in the terrain,

the likelihood of associated avalanches and the destructive size of these avalanches. The CMAH defines nine different avalanche problem types, which represent typical, repeatable patterns of avalanche hazard formation and evolution. Identifying the type of an avalanche problem is a critically important step in the hazard assessment process as it provides an overarching filter that sets expectations and influences subsequent decisions about relevant types of observation and effective approaches for risk reduction. The broad adoption of the CMAH among North American avalanche safety

practitioners and public avalanche forecasters opens new opportunities for including avalanche bulletin information in formal research (see, e.g., Shandro and Haegeli, 2018).

## 3 Method

### 3.1 Avalanche bulletin data

The foundation for the present study is CMAH-compliant avalanche hazard assessments included in daily public avalanche

bulletins published by Avalanche Canada, Parks Canada and Alberta Parks during the winter seasons 2010 to 2019.[1] Together, the three agencies provide daily avalanche forecasts for all main mountain ranges in western Canada, which include the maritime Coast Mountains along the Pacific Coast in the west, the continental Rocky Mountains along the BC-

---

[1] Winter seasons are labelled with the year when the winter finishes. Hence 2010 represents the winter from Dec. 2009 to Apr. 2010.

Alberta boarder in the east, and the Columbia Mountains that exhibit a transitional snow climate in between (McClung and Schaerer, 2006) (Figure 1). During the first two winters (2010 and 2011), the dataset is limited to six large forecast areas of

Avalanche Canada (Northwest-BC, South Coast, North Columbia, South Columbia, Kootenay Boundary and South Rockies). In the winter 2012, most of these regions were subdivided into subregions to provide recreationists with more location-specific avalanche hazard information. In the same season, Parks Canada and Alberta Parks implemented the use of the CMAH as the foundation for their avalanche bulletins. This means that for 2012 to 2019 winter seasons our data consists of daily avalanche hazard analyses from 15 different forecast areas (Figure 1). To increase consistency among forecast areas

and winters, we only included bulletins that were published between Dec. 1 and Apr. 15. The number of avalanche bulletins per season and forecast area is typically 136 or 137, except during the first two winters when the number was slightly smaller (range: 92 to 131).

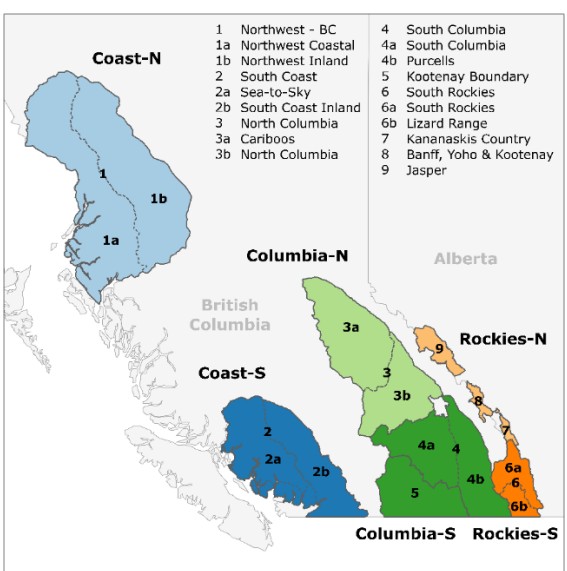

**Figure 1: Overview of study area with avalanche forecast areas and analysis regions. Labels of forecast areas express the relationship between the large forecast areas from the first two winters and the smaller forecast areas thereafter (e.g., Northwest-BC (1) to split into Northwest Coastal (1a) and Northwest Inland (1b)).**

For the present analysis, we grouped the forecast areas into six large-scale regions: *Coast-North*, *Coast-South*, *Columbias-*

*North*, *Columbias-South*, *Rockies-North* and *Rockies-South* (Figure 1). The Glacier National Park forecast area was excluded from the analysis as it is a small forecast area that is located right between Columbias-North and Columbias-South. Furthermore, it is the only Parks Canada forecast area in the Columbia Mountains and their daily schedule for publishing the avalanche bulletin is different from all the other areas. The complete avalanche bulletin dataset consisted of 16,867 daily avalanche bulletins over ten seasons from 15 forecast areas grouped into six large-scale regions. Organizing the forecast

areas into large-scale regions has several advantages for our analysis. First, it allows us to include the complete dataset in the

analysis despite the splitting of some of the forecast regions after the first two winters. Second, it strengthens the relatively short dataset by including multiple observations per region, and third, it helps to smooth out small-scale variabilities that might be artifacts of the short dataset and difficult to interpret.

Our analysis focused on the 'Day zero' avalanche hazard assessment that avalanche forecasters make for the current day based on all available information before they produce hazard forecasts for the upcoming days. To prepare the hazard assessments for the present analysis, we calculated fractions of forecast days when a specific avalanche problem type was present for each season, elevation band (alpine, treeline, and below treeline) and forecast region. This means that each winter season for a forecast area and elevation band is characterized by a set of eight percentage values, one for each avalanche problem type (*Storm slab avalanche problems*, *Wind slab avalanche problems*, *Persistent slab avalanche problems*, *Deep persistent slab avalanche problem*, *Wet slab avalanche problem*, *Wet loose avalanche problem*, *Dry loose avalanche problem*, and *Cornice avalanche problem*). In addition, we also computed the fractions of days when *No avalanche problems* were present and the fractions of days with persistent or deep persistent slab avalanche problems as forecasters have expressed challenges with reliably distinguishing these two avalanche problems types (Grant Statham: personal communication). While the avalanche hazard characterization method developed by Shandro and Haegeli (2018) provides a more integrated perspective of conditions that also includes the severity of the conditions, we chose the simpler approach of focusing on the prevalence of individual avalanche problem types to make the results easier to interpret and simplifying the steps for reproducing the approach in other geographic regions.

The prevalence values included in our dataset vary considerably among avalanche problem types, forecast areas, season and elevation bands (Table 1 and Figure 2). During our study period *Storm slab avalanche problems*, *Wind slab avalanche problems*, and *Persistent slab avalanche problems* were the predominant avalanche problems in the alpine and treeline elevation bands. The most prevalent avalanche problems below treeline were *No avalanche problems*, *Storm slab avalanche problems*, and *Persistent slab avalanche problems*, whereas *Wind slab avalanche problems* and *Cornice avalanche problems* were rare.

**Table 1: Avalanche problem types (Statham et al., 2018) and summary of seasonal prevalence values (fractions of forecast days when a specific avalanche problem type was present per season from Dec. 1 to Apr. 15) for the three elevation bands alpine (ALP), treeline (TL) and below treeline (BTL).**

| Avalanche problem type | Description (Statham et al., 2018) | Seasonal prevalence values (median \| max.) | | |
|---|---|---|---|---|
| | | ALP | TL | BTL |
| a) Storm slab avalanche problem | Cohesive slab of soft new snow. Also called a direct-action avalanche. | 35 \| 65 | 36 \| 62 | 25 \| 53 |
| b) Wind slab avalanche problem | Cohesive slab of locally deep, wind-deposited snow. | 57 \| 98 | 50 \| 96 | 2 \| 19 |
| c) Persistent slab avalanche problem | Cohesive slab of old and/or new snow that is poorly bonded to a persistent weak layer and does not strengthen or strengthens slowly over time. Structure is conducive to failure initiation and crack propagation. | 37 \| 88 | 43 \| 88 | 26 \| 67 |
| d) Deep persistent slab avalanche problem | Thick, hard cohesive slab of old snow overlying an early-season persistent weak layer located in the lower snowpack or near the ground. Structure is conducive to failure initiation and crack propagation. Typically characterized by low likelihood and large destructive size. | 10 \| 97 | 10 \| 91 | 0 \| 50 |
| e) All persistent slab avalanche problem | Combines persistent and deep persistent slab avalanche problems. | 57 \| 100 | 63 \| 100 | 29 \| 68 |
| f) Wet slab avalanche problem | Cohesive slab of moist to wet snow that results in dense debris with no powder cloud. | 1 \| 7 | 0 \| 8 | 0 \| 14 |
| g) Wet loose avalanche problem | Cohesionless wet snow starting from a point. Also called a sluff or point release. | 10 \| 27 | 13 \| 33 | 13 \| 38 |
| h) Dry loose avalanche problem | Cohesionless dry snow starting from a point. Also called a sluff or point release. | 5 \| 34 | 3 \| 29 | 2 \| 25 |
| i) Cornice avalanche problem | Overhanging mass of dense, wind deposited snow jutting out over a drop-off in the terrain. | 14 \| 64 | 0 \| 40 | 0 \| 0 |
| j) No avalanche problem | Situations when no avalanche problem is present. | 1 \| 9 | 6 \| 21 | 37 \| 83 |

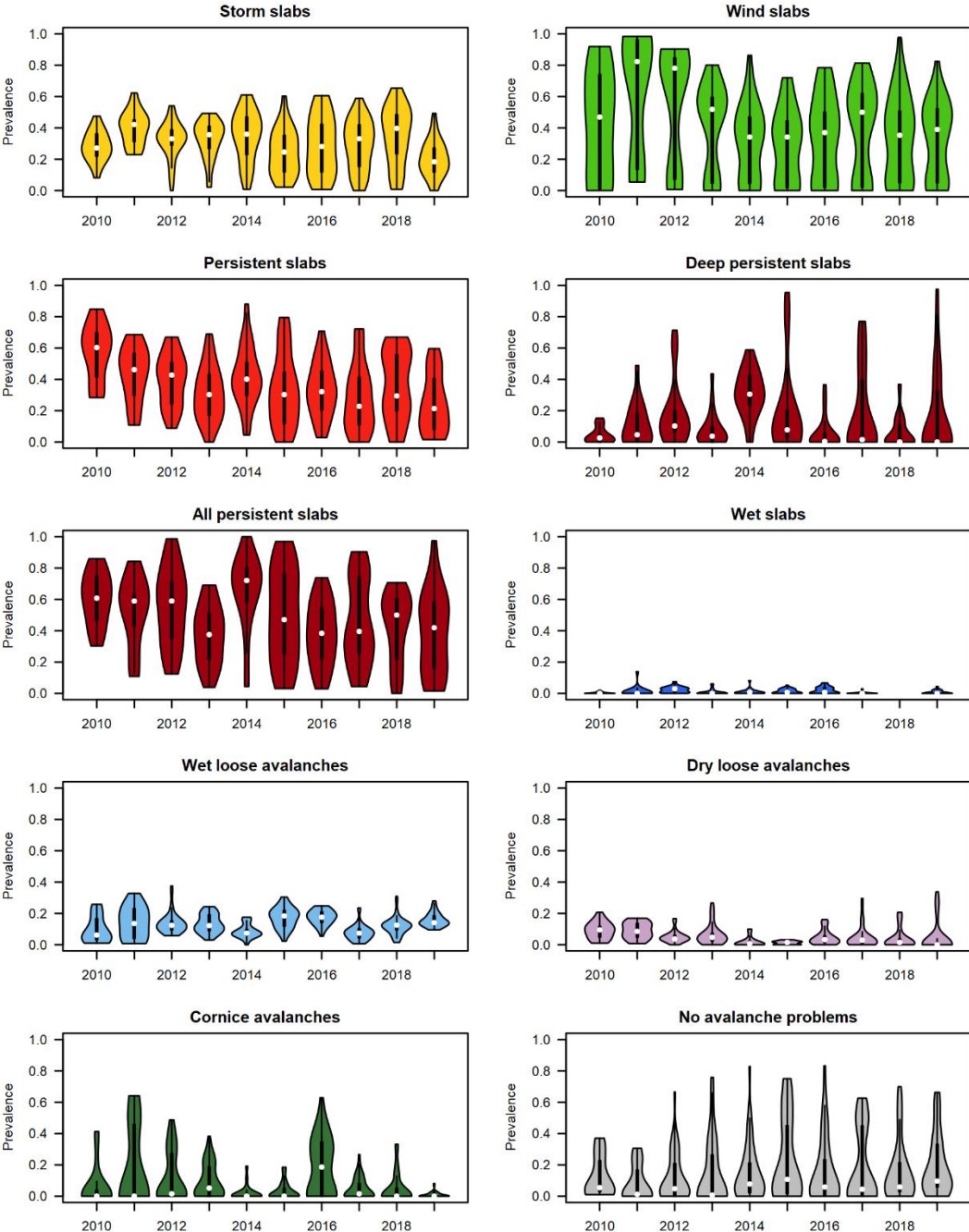

**Figure 2: Time series of violin plots illustrating changes in the seasonal distributions of prevalence values for each avalanche problem type. Each violin plot represents the observed prevalence values from all regions and elevation bands (18 observations in 2010 and 2011; 45 observations per winter thereafter). White dots in violin plots represent median and thick black lines show interquartile ranges.**

## 3.2 Information on atmosphere-ocean oscillations

We used publicly available data from the National Oceanic and Atmospheric Administration (NOAA) of the U.S. Department of Commerce for characterizing the various atmosphere-ocean oscillations. Various indices are used to identify the phase and describe the strength of ENSO. In this study, we used the revised version of the Multivariate El Niño Index (MEI.v2; Wolter and Timlin, 2011; Zhang et al., 2019), which considers five main parameters observed over the tropical Pacific, including sea-level pressure, surface zonal and meridional winds, sea surface temperature, and outgoing longwave

radiation for calculating the strength of ENSO. Bimonthly MEI.v2 values can be downloaded from the website of NOAA's Physical Science Laboratory (2020). The intensity of the PDO is described with the PDO index, which is calculated from monthly sea surface temperature anomalies and the monthly mean global average sea surface temperature anomaly (Mantua et al., 1997). The PNA is measured with the PNA index, which relates to anomalies in the 700 mb and 500 mb geopotential height fields observed over Western and Eastern North America (Zhao et al., 2013), with mean flow characterized by a

trough in the Eastern-Central Pacific, and a ridge over the Rocky Mountains (Whitfield et al., 2010). The AO is described with the AO index (Thompson and Wallace, 1998), which incorporates non-seasonal sea-level pressure variations north of 20-degree latitude. We downloaded monthly values of the PDO, PNA, and AO indices from the website of NOAA's National Centers for Environmental Information (2020).

Following established practices in hydrological studies on the effect of atmosphere-ocean oscillations (e.g., Fleming and

Dahlke, 2014a), we calculated seasonal indices for the strength of the individual atmosphere-climate oscillations by averaging the values of the winter months (MEI.v2: Nov./Dec. to Mar./Apr.; PNA, PDA and AO: Nov. to Apr.) for each winter between 2010 and 2019. While the study period is limited to ten years, all four climate indices exhibited both negative and positive anomalies and covered between 64 % and 84 % of the historical range (Table 2). Our study period includes ENSO observations near the historical minimum (2011), and the AO index exhibited its historical minimum in the winter of

235 2010.

Since the resulting seasonal indices for the Pacific-centered atmosphere-ocean oscillations were highly correlated (Figure 2; MEI.v2 vs PDO: 0.71 (Pearson correlation coefficient); MEI.v2 vs PNA: 0.83; PDO vs PNA: 0.54) it would not be possible for our analysis to isolate their individual effects in a meaningful way. To properly include the effect of these atmosphere-ocean oscillations in our analysis and prevent inappropriate conclusions, we calculated a seasonal climate oscillation index

for the combined strength of the Pacific-centered oscillations (POs) by averaging the ENSO, PDO and PNA indices for each winter (Figure 3). The time series of the seasonal AO index is distinctly different from the Pacific-centered oscillations (Pearson correlations ranging between -0.24 and 0.20) and its correlation with the combined POs index was only 0.05. This is consistent with the independence between PO and AO described in previous studies (e.g., Wu and Hsieh, 2004) and ideal for separating the effects of the two types of atmosphere-ocean oscillations in the analysis.


**Table 2: Overview of monthly atmosphere-ocean oscillation indices (ENSO: El Nino-Southern Oscillation; PNA: Pacific North America Teleconnection Pattern; PDO: Pacific Decadal Oscillation; AO: Arctic Oscillation, PO: Averaged Pacific-centered oscillations).**

| Index | Historical range* | | Observations during study period (2010-2019) | | |
|---|---|---|---|---|---|
| | Min | Max | Min | Max | Percentage of hist. range |
| ENSO (MEI.v2) | -2.43 | 2.89 | -2.04 | 1.94 | 75% |
| PNA index | -3.07 | 2.66 | -2.06 | 2.02 | 71% |
| PDO index | -3.65 | 3.84 | -2.95 | 1.85 | 64% |
| AO index | -4.27 | 3.50 | -4.27 | 2.80 | 84% |

\* MEI.v2: January 1979 to April 2019; PNA, PDO and AO: January 1950 to April 2019.


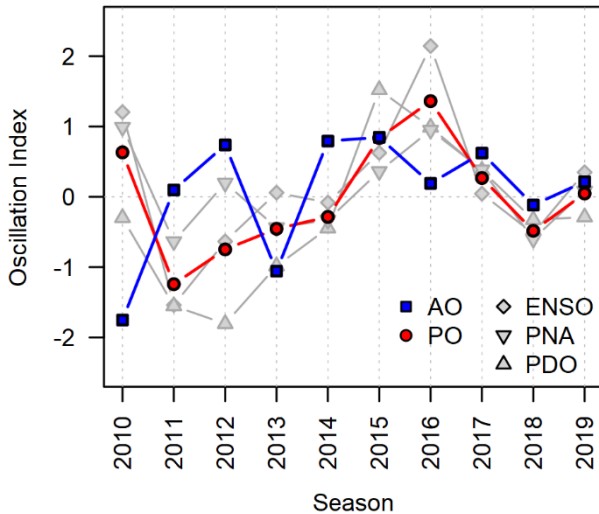

**Figure 3: Winter season (Nov. – Apr.) average climate indices during the study period (ENSO: El Nino-Southern Oscillation; PNA: Pacific North America Teleconnection Pattern; PDO: Pacific Decadal Oscillation; AO: Arctic Oscillation, PO: Averaged Pacific-centered oscillations).**


### 3.3 Statistical analysis

While a 10-year dataset is relatively short for a climatological study, our analysis approach aims to maximize the value of

the available data to provide meaningful insight into the relationship between the combined Pacific-centered atmospheric-ocean oscillations and the AO and the nature of avalanche hazard in western Canada at the regional scale. Whereas previous studies employed correlation analyses to explore these relationships one at a time, we used generalized linear mixed effects regression models to simultaneously examine and properly isolate the effect of the two different types of oscillations. Since our dependent variable are prevalence values that are bound between 0 and 1 and considerably skewed towards lower values,

we chose beta regression models (Cribari-Neto and Zeileis, 2010; Smithson and Verkuilen, 2006) with a logit link function for our analysis (see Appendix for formal expression of model). As suggested by Smithson and Verkuilen (2006), we transformed our prevalence values with $y_{trans} = (y_{orig}(n-1) + 0.5)/n$ prior to analysis to eliminate values that are exactly zero or one since they cannot be handled by the beta regression.

It is well known that the indices of atmospheric oscillations like the PDO or AO exhibit considerable autocorrelations.

Newman et al. (2016), for example, point out that the year-to-year PDO correlation is over 0.45 in late winter and spring. However, since the seasonal snowpack in Western Canada largely melts out every summer, and the snowpack structures relevant for avalanches emerge each winter independently of the previous winter, it is not necessary to use an autoregressive model approach for the present analysis.

We estimated separate mixed effects models for each avalanche problem type. Each of these models included the

atmosphere-ocean oscillation indices (POs and AO) and the large-scale regions as fixed effects. Winter season was included in the models as a random effect to account for the intricacies of individual winters that cannot be explained by the atmosphere-ocean oscillations included in the analysis. Due to the stronger similarity in the prevalence of avalanche problems between the alpine and treeline elevation bands (Table 3), we combined the analysis of the two elevation bands and estimated single models for prevalence values in the two elevation bands with elevation band as an additional fixed effect.

The models for below treeline were estimated separately. We did not estimate a below treeline model for *Cornice avalanche problems* because this avalanche problem is not relevant at lower elevations. Hence, we conducted 19 different regression model analyses in total.

**Table 3: Pearson correlation coefficients for prevalence of avalanche problem types between different elevation bands (ALP: alpine; TL: treeline; and BTL: below treeline).**

| | Storm slabs | Wind slabs | Persistent slabs | Deep persistent slabs | Wet slabs | Wet loose aval. | Dry loose aval. | Cornice aval. | No aval. prob. |
|---|---|---|---|---|---|---|---|---|---|
| ALP – TL | 0.97 | 0.92 | 0.89 | 0.98 | 0.83 | 0.83 | 0.93 | 0.55 | 0.52 |
| ALP – BTL | 0.82 | 0.33 | 0.56 | 0.65 | 0.36 | 0.44 | 0.81 | 0.03 | -0.11 |
| TL – BTL | 0.87 | 0.42 | 0.74 | 0.74 | 0.60 | 0.76 | 0.92 | 0.02 | 0.34 |


To explore the spatial patterns in the effect of the atmospheric oscillations, each of these analyses included two model estimations. We first estimated a simple model that only included AO, PO and large-scale region as main effects. All of the categorical variables were effects coded, so that the parameter estimates for large-scale regions capture the average differences in the prevalence of the specific avalanche problem type across the entire study period, and the parameter estimates for AO and PO describe the average effect of the atmospheric oscillations across the entire study area. Our second model also included interactions between atmospheric oscillation variables and the large-scale regions to resolve potential spatial differences in the response to AO and PO. We then used a likelihood ratio test to determine whether the second and more complex model with interactions represented the data better than the simpler main effects model. We picked the interaction model as the final model if the p-value of the likelihood ratio test was below 0.05, and we stayed with the simpler main effects model if it was not.

Our preliminary analysis of the prevalence data indicated an abnormally high prevalence of *Wind slab avalanche problems* in the first three seasons (Figure 2: 2010, 2011 and 2012). A closer examination revealed that this anomaly is likely related to conditions when avalanche forecasters were simultaneously concerned about storm and wind slab avalanches. The analysis of Shandro and Haegeli (2018) explicitly identified these types of hazard situations and labelled them as Storm & wind slab and Storm, wind & persistent slab hazard situations. To make the avalanche problem information in their bulletins more distinct, Avalanche Canada instituted a new internal forecasting policy at the beginning of the 2013 winter season that discourages forecasters from including storm and wind slabs in the same forecasts (Shandro and Haegeli, 2018). To account for this change in forecasting practice in our analysis, we included an additional binary variable in our dataset that was set to 1 for Avalanche Canada for the first three seasons (2010, 2011 and 2012) and 0 otherwise. We then integrated the variable as an additional fixed effect in the models for *Storm slab* and *Wind slab avalanche problems* under the assumption that the policy change may be associated with a consistent change in the prevalence values across all Avalanche Canada forecast regions.

We conducted our entire analysis in R (R Core Team, 2020) and used the `glmmTMB` package (Brooks et al., 2017) to estimate our mixed effects models. Because of the relatively small dataset, we did not only consider parameter estimates with p-values < 0.05, but also viewed parameter estimates with p-values between 0.05 and 0.10 to be indicative of marginally significant trends. To assess violations in model assumptions, we simulated quantile residuals (Dunn and Smyth, 1996) as implemented in the `DHARMa` package (Hartig, 2020). Visual inspection of the resulting diagnostic plots (e.g., Q-Q-plot for uniformly distributed residuals) did not suggest any substantial model violations. Due to the logit link function of the beta regression, the parameter estimates are difficult to interpret directly and converting them into odds ratios does not simplify the interpretation as they represent odds of percentages. In addition, making sense of the combined main and interaction effects is particularly challenging in logistic regressions. To make the interpretation of the results more tangible, we used the parameter estimates from the regression analyses for the different avalanche problems to calculate their expected prevalence values across the value ranges of the AO and PO indices that were observed during the study period. We then followed up with post-hoc pairwise comparisons to assess whether the marginal mean estimates (i.e., the mean estimates of the

prevalence values at the minimum and maximum values of the AO and PO indices) were significantly different from each other for the different large-scale regions. In other words, we tested whether the change in the prevalence of an avalanche problem expressed in percentage points was significantly different from zero. We performed this part of the analysis using the `emmeans` and `pairs` functions of the `emmeans` package (Lenth, 2019). To counteract the issue of Type I error inflation from multiple comparisons, we calculated Holm-corrected p-values.

## 4 Results

Our presentation of the results focuses on the relationship between the atmosphere-ocean oscillations and the nature of avalanche hazard in western Canada at the regional scale. We therefore concentrate on the examination of the main effects of AO and PO as well as their interactions with the large-scale region. The main effect of large-scale region and the random intercept for winter season are not discussed because they only reflect the regional and seasonal variability in the average
prevalence of avalanche problem types respectively. Interested readers are referred to Shandro and Haegeli (2018) for a detailed description of these types of variabilities.

The results of our analysis are summarized in Figure 4, which shows the effect of AO and PO on the prevalence values of individual avalanche problem types expressed as changes in percentage points over the range of the observed oscillation indices (i.e., difference in marginal mean estimates). For each avalanche problem type and elevation band (alpine/treeline
and below treeline) the six percentage point values are arranged to roughly represent the geographic arrangement of the large-scale regions (Figure 1). To provide a more in-depth perspective on the relationship between the atmospheric oscillations and the prevalence values, effects plots are used for select avalanche problem types of interest (Figures 5-8). These plots show the logistic relationships between the mean prevalence value and the AO or PO indices together with the 95 % confidence interval for the different large-scale regions. The individual points in the figures represent observed
prevalence values.

Our presentation of the results focuses primarily on the big picture patterns that emerged from the analysis and does not discuss each model in detail. However, interested readers are referred to the available data and analysis code for detailed information on the parameter estimates of the final models for each of the avalanche problem types and elevation bands. When interpreting the percentage point changes in Figure 4, it is important to realize that the presented values are a
combination of both the magnitude of the effect of the atmospheric oscillation (i.e., the size of the regression parameters) and the average prevalence of the avalanche problem in the region over the study period. This means that the same effect will produce smaller percentage point changes in regions with lower average prevalence values of the avalanche problem and larger values in regions with higher prevalence values.

Overall, seven of the ten models for the alpine/treeline elevations included interactions effects for region whereas none of the
nine below treeline models did. A possible explanation of this result is that the below treeline response to atmospheric oscillations is more homogeneous across the entire study area than at higher elevations because the warmer temperatures at

lower elevations mean that the snowpack is generally closer to the melting point and therefore more sensitive to temperature variations. However, one also needs to remember that we can expect higher levels of significance in the alpine/treeline models since the available number of observations for those models is twice as large as for the below treeline model. Hence the differences in the spatial patterns across elevation bands should be treated with caution.

Based on the results of the models, the relationships between the prevalence of avalanche problem types and the atmosphere-ocean oscillations can be grouped into four classes. The prevalence values of the problem types are either a) not affected by the atmosphere-ocean oscillations (i.e., no significant main or interaction effects), b) respond consistently across the entire study area without detectable regional variability (only significant main effects); c) respond regionally different in addition to the overall study area wide effect (significant main and interaction effects); or d) respond regionally different without an overall effect across the entire study area (only significant interaction effect). The effect of PO emerged as a consistent pattern across the study area in five (26 %) of the 19 models, as a consistent pattern across the study area with regional differences in only one (5 %) model, and regional differences only in three models (16 %) (Figure 4). No effect was observed in ten models (52 %). With respect to AO, eight of the 19 models (42 %) did not exhibit an effect at all, four (21 %) had a consistent effect across the entire study area, in two models (11 %) the consistent effect across the study area was superimposed with regional differences, and the remaining five models (26 %) exhibited a regional pattern without a consistent effect across the study area. These results clearly highlight that not all avalanche problem types are affected by the atmospheric oscillations, and that the response can vary regionally considerably. This is consistent with the results of several studies that have shown considerable regional differences in the weather patterns related to atmospheric oscillation anomalies in western Canada and the Pacific Northwest (e.g., Jin et al., 2006; Wise, 2010; Fleming and Whitfield, 2010).

The following sections provide an overview of the observed effects of PO and AO on the nature of avalanche hazard in the two elevation bands alpine/treeline and below treeline. We focus on the big picture patterns and illustrate regional differences with a few examples.

**Avalanche Problem Types**

| | ARCTIC OSCILLATION (-1.75 to 0.84) | | PACIFIC OSCILLATIONS (-1.24 to 1.36) | |
|---|---|---|---|---|
| | Alpine/Treeline | Below treeline | Alpine/Treeline | Below treeline |

**a) Storm slabs**

| 14 | 4 | -8 | | -1 | -1 | 0 | | -13 | -8 | **-15** | | **-17** | **-19** | **-4** |
| -6 | -5 | -2 | | -1 | -1 | 0 | | -11 | -9 | **-23** | | **-18** | **-18** | **-15** |
| | IA only | | | | | | ME only | | | ME only | |

**b) Wind slabs**

| -7 | 9 | 15 | | 3 | 1 | 1 | | -8 | **-21** | **-17** | | **-11** | **-5** | **-3** |
| -4 | **15** | 6 | | 1 | 2 | 2 | | **-16** | **-26** | -6 | | **-4** | **-6** | **-6** |
| | IA only | | | ME only | | | ME and IA | | | ME only | |

**c) Persistent slabs**

| -11 | -11 | -11 | | -9 | -12 | -5 | | -2 | -2 | -2 | | -3 | -4 | -2 |
| -8 | -11 | -11 | | -4 | -12 | -9 | | -2 | -2 | -2 | | -1 | -4 | -3 |

**d) Deep pers. slab**

| -7 | 7 | **50** | | **4** | **5** | **8** | | 6 | -12 | 11 | | -2 | -2 | -4 |
| 1 | 10 | **24** | | **3** | **5** | **9** | | -7 | -9 | **-16** | | -2 | -3 | -4 |
| | ME and IA | | | ME only | | | IA only | | | | | |

**e) All pers. slabs**

| -19 | 3 | **56** | | -3 | -4 | -3 | | 8 | -8 | -14 | | -6 | -8 | -6 |
| 5 | -6 | 23 | | -2 | -4 | -4 | | -8 | -7 | -20 | | -3 | -8 | -7 |
| | IA only | | | | | | | | | | | |

**f) Wet slabs**

| **1** | 1 | -1 | | 2 | 2 | 1 | | -1 | 1 | 0 | | -1 | -1 | 0 |
| 1 | 1 | **2** | | 1 | 2 | 1 | | **1** | **1** | **3** | | -1 | -1 | -1 |
| | IA only | | | ME only | | | IA only | | | | | |

**g) Wet loose avalanches**

| 7 | **8** | 6 | | -3 | -3 | -3 | | 0 | 2 | 1 | | 1 | 1 | 1 |
| 7 | 6 | -5 | | -4 | -3 | -3 | | 7 | 6 | **11** | | 1 | 1 | 1 |
| | IA only | | | | | | | IA only | | | | | |

**h) Dry loose avalanches**

| **-9** | -4 | **-42** | | -2 | **-5** | **-11** | | -4 | 3 | -7 | | -3 | **-6** | **-13** |
| -8 | -6 | **-8** | | -3 | -4 | -4 | | -4 | -1 | -7 | | **-4** | **-5** | **-5** |
| | ME and IA | | | ME only | | | | | | ME only | |

**i) Cornices**

| -2 | -3 | -2 | | | Not relevant | | | 1 | 2 | 1 | | | Not relevant | |
| -4 | -3 | -4 | | | | | | 2 | 2 | 2 | | | | |

**i) No problems**

| 1 | 1 | 1 | | 5 | 5 | 5 | | 2 | 2 | 2 | | **22** | **19** | **23** |
| 2 | 1 | 1 | | 5 | 5 | 5 | | 3 | 2 | 2 | | **23** | **19** | **22** |
| | | | | | | | | | | | ME only | |

**Legends**

Large-scale regions:

| CoastN | ColN | RockN |
|---|---|---|
| CoastS | ColS | RockS |

Magnitude of change:

| > 10 | 5 - 10 | 0 - 5 |
|---|---|---|
| < -10 | -10 - -5 | -5 - 0 |

Statistical significance:
- **p-value < 0.05**
- p-value < 0.10
- not statistically signifcant

**Figure 4: Overview of the effect of the examined atmospheric oscillations on the prevalence values of avalanche problem types expressed as change in percentage points over the range of the observed oscillation indices (i.e., difference in marginal mean estimates). For each avalanche problem type and elevation band (alpine/treeline and below treeline) the six percentage point values are arranged to roughly represent their geographic arrangement. Font weight and color of the percentage point labels describe the statistical significance of the difference: bold and black/white: p < 0.05; black/white only: 0.05 ≤ p < 0.10; grey: not statistically significant. The shading of the cell indicates the magnitude of any significant percentage point differences: dark blue: < -10 pp; medium blue: -10 to -5 pp; faint blue: -5 to 0 pp; faint red: 0 to 5 pp; medium red: 5 to 10 pp; dark red: > 10 pp. The labels underneath the boxes indicate whether the model includes significant the main (ME) and/or interaction effects (IA).**

## 4.1 Response to Pacific-centered oscillation

One of the prominent patterns in our results is the strong and uniform negative association between PO and the prevalence of *Storm slab* and *Dry loose avalanche problems* below treeline (Figure 4 and 5). Both of these avalanche problem types are less prevalent during the positive phase of the oscillation and more prevalent during the negative phase. Complementary to this pattern, we observed a significant positive relationship between PO and the prevalence of days with *No avalanche problems* (Figure 4 and 6). These observations are consistent with the existing understanding of the effect of PO on the winter weather in the southern parts of BC and the Pacific Northwest as the warmer temperatures experienced during the positive phase (Shabbar and Khandekar, 1996; Shabbar and Bonsal, 2004; Bonsal et al., 2001) generally result in a shallower and less hazardous snowpack at lower elevations. The observed pattern is also consistent with the results of Lute and Abatzoglou (2014), who showed that La Niña winters in the Pacific Northwest are generally associated with above normal snow water equivalents that result from both more snowfall days and more extreme snow fall events compared to El Niño winters, and the studies of Brown and Goodison (1996) and Moore and McKendry (1996) who showed that the positive phases of both ENSO and PNA are associated with reduced snow cover western Canada. Hence, our prevalence values for alpine/treeline *Storm slab avalanche problems* exhibit the expected negative association with PO at higher elevations (Figure 4 and 5). Consistent with the previous research, our regression analysis indicates a homogeneous effect of PO across the study area (i.e., no significant interaction effect), but the magnitude of the estimated difference over the observed PO index is most pronounced in the Rocky Mountains. While Fleming and Whitfield (2010) point out that the northern coast of BC and Alaska exhibits an inverse response pattern for precipitation with the warm ENSO phase bringing wetter winter and spring conditions, this deviation would only affect the Coast North region of our study area.

Another interesting widespread pattern is the also negative relationship between PO and the prevalence of *Wind slab avalanche problems* across the study area (Figure 4 and 7). In the alpine/treeline elevation band, the pattern is a combination of an overall negative effect across the study area that is further enhanced by a negative interaction effect in the Columbia-South region. Combined, the magnitude of the estimated difference over the observed PO index is largest in the Columbia Mountains followed by the Rockies-North and Coast-South regions. The reduction in *Wind slab avalanche problems* is also observed below treeline, but it is important to remember that this type of avalanche problem is only rarely an issue at lower elevations (Figure 7, bottom row). The observed change in the prevalence of *Wind slab avalanche problems* can potentially be explained with the changes in the large-scale circulation patterns associated with the PNA. As described by Bonsal et al. (2001) and Stahl et al. (2006), the enhanced ridge over western North America during the positive phase of the PNA results in a weaker and more meridional flow pattern over the study area than the more zonal flow pattern during the negative phase.

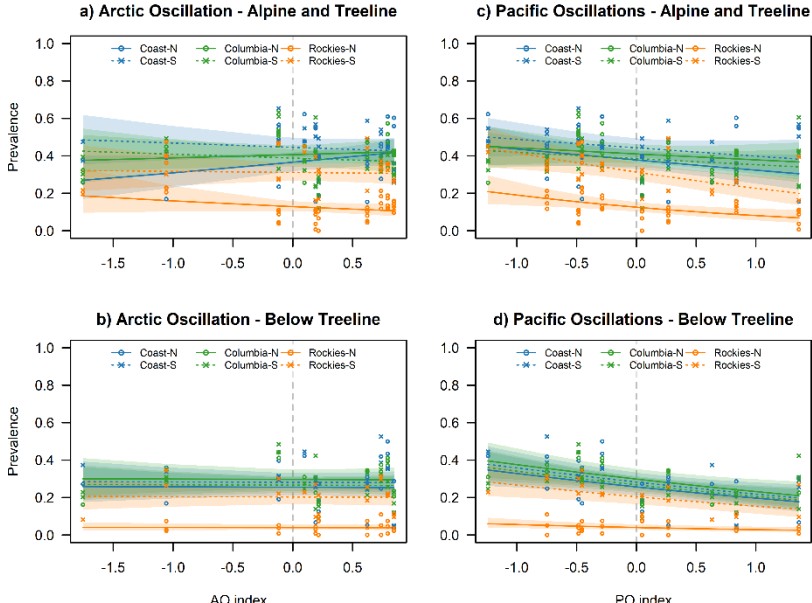

**Figure 5: Calculated prevalence for *Storm slab avalanche problems* in relation to AO (left column) and Pacific-centered oscillations (right column) for the alpine/treeline (top row) and below treeline (bottom row) over the range of observed index values. Lines indicate the mean estimates and shaded areas represent that 95% confidence intervals. Individual points represent observed prevalence values.**

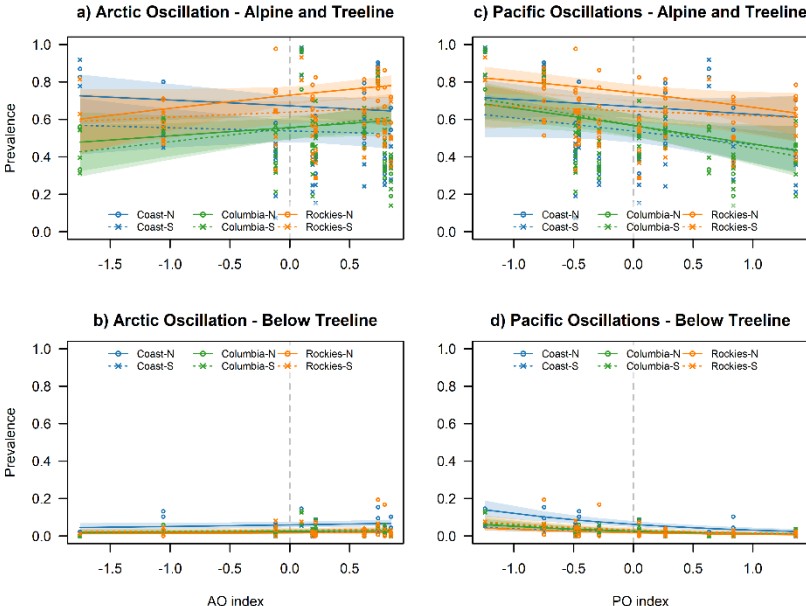

**Figure 6: Calculated prevalence for *Wind slab avalanche problems* in relation to AO (left column) and Pacific-centered oscillations (right column) for the alpine/treeline (top row) and below treeline (bottom row) over the range of observed index values. Lines indicate the mean estimates and shaded areas represent that 95% confidence intervals. Individual points represent observed prevalence values.**

Furthermore, the enhanced ridge and associated northern displacement of the jet stream during the positive phase also inhibits the formation Arctic outflow situations (Bonsal et al., 2001). Both of these effects together offer a reasonable explanation for the observed pattern in the prevalence of *Wind slab avalanche problems*.

When interpreting the prevalence of *Strom slab* and *Wind slab avalanche problems*, it is important to remember the change in forecasting practice at Avalanche Canada at the beginning of the 2013 winter season. The additional variable included in the model to account for this change was only marginally significant for *Storm slab avalanche problems* in the alpine/treeline model (-0.337; p-value = 0.093), but highly significant for *Wind slab avalanche problems* in the alpine/treeline model (1.540; p-value < 0.001). This indicates that Avalanche Canada forecasters included *Wind slab avalanche problems* substantially more frequently in the hazard assessments and *Storm slab avalanche problems* slightly less frequently before the practice change. Having explicitly accounted for this change in forecasting practices, we can be more confident that the identified changes in the prevalence of *Wind slab avalanche problems* are associated with PO. In addition to the large-scale patterns described above, we also observe several more regional patterns. First, we see a positive relationship between PO and the prevalence of *Wet slab avalanche problems* in the southern regions of the study area (Figure 4). While the absolute change is relatively small, it is partially due to the fact that wet slabs are generally forecasted rarely (Figure 2). This observation is consistent with the results of McClung (2013) and Thumlert et al. (2014) who describe positive associatations between the percentage of wet-snow avalanches and ENSO and PDO, but the effect in our study is substantially smaller. This discrepancy is likely explained by the fact that McClung (2013) and Thumlert et al. (2014) defined wet avalanches based on the recorded liquid water content of the avalanche deposit (Canadian Avalanche Association, 2016: dry, moist or wet). This means that their percentage of wet avalanches also includes avalanches that started dry but became wet as they reached lower elevations. A second potential reason for the lower prevalence of wet avalanches in our study is that we limited our datasets to between December 1 and April 15, which likely prevents widespread wet avalanche cycles in the spring to be included.

A second set of regional PO response pattern observed in our results include a negative relationship with the prevalence of *Deep persistent slab avalanche problems* and a positive relationship with *Wet loose avalanche problems* in the alpine/treeline models in the Rockies-South region (Figure 4). Both patterns are potentially consistent with the higher temperatures during positive PO phases. The isolated response of the Rockies-South region is not overly surprising as the southeast corner of British Columbia is well known for being exposed to different weather systems and having a unique snow climate that is distinct from the surrounding areas. While the more northern parts of the Canadian Rocky Mountains exhibit a traditional continental snow climate, the southern parts have a more transitional snow climate with warmer temperatures and a deeper snowpack (Claus et al., 1984; Johnston, 2011; Haegeli and McClung, 2007).

Interestingly, our analysis did not reveal a substantial relationship between PO and the prevalence of *Persistent* or *Deep persistent slab avalanche problems*, expect the local effect on *Deep persistent slab avalanche problems* in the Rockies-South region. To provide context for the interpretation of this result, it is also important to remember that the logit link of the beta regression can only capture monotonic relationships between the prevalence of an avalanche problem type and the oscillation

indices. This may be an issue for the analysis of *Persistent slab avalanche problems* that are most common in transitional snow climates where we have a combination of both maritime and continental influences (Haegeli and McClung, 2007).

Analogously, neutral atmosphere-ocean oscillation conditions might be most favorable for this type of avalanche problem. Non-monotonic response patterns to PO in western Canada have also been identified in hydrological studies such as Fleming and Dahlke (2014a), Fleming and Dahlke (2014b) and Fleming et al. (2016).

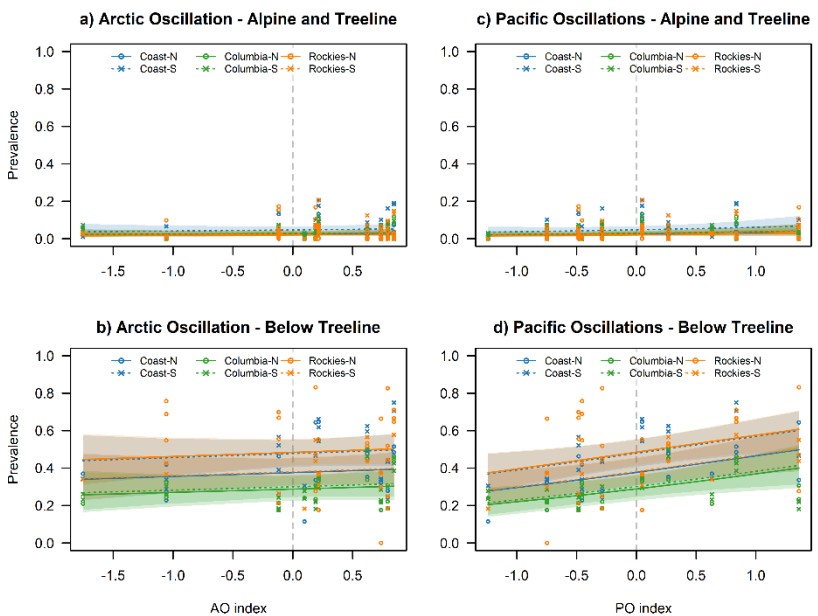

**Figure 7: Calculated prevalence for *No avalanche problems* in relation to AO (left column) and Pacific-centered oscillations (right column) for the alpine/treeline (top row) and below treeline (bottom row) over the range of observed index values. Lines indicate the mean estimates and shaded areas represent that 95% confidence intervals. Individual points represent observed prevalence values.**

**4.2 Response to Arctic Oscillation**

One of the prominent AO response patterns in our analysis is the increase in the prevalence of *Deep persistent slab avalanche problems* across a substantial part of the study area (Figure 4 and 8). While the main effect in the alpine/treeline model is relatively weak (0.386; p-value = 0.056), significant interactions describe a stronger effect in the Rocky Mountains and a diminished effect in the Coast Mountains. The pattern is more uniform in the below treeline model. However, it is

475 important to remember that the large change in the prevalence in the Rocky Mountains is a combination of the positive interactions as well as the higher prevalence of *Deep persistent slab avalanche problems* in the continental snow climate in general (Figure 8 and Shandro and Haegeli, 2018). Similar to the response to the PO, we did not identify a significant effect of AO on the prevalence of *Persistent slab avalanche problems*.

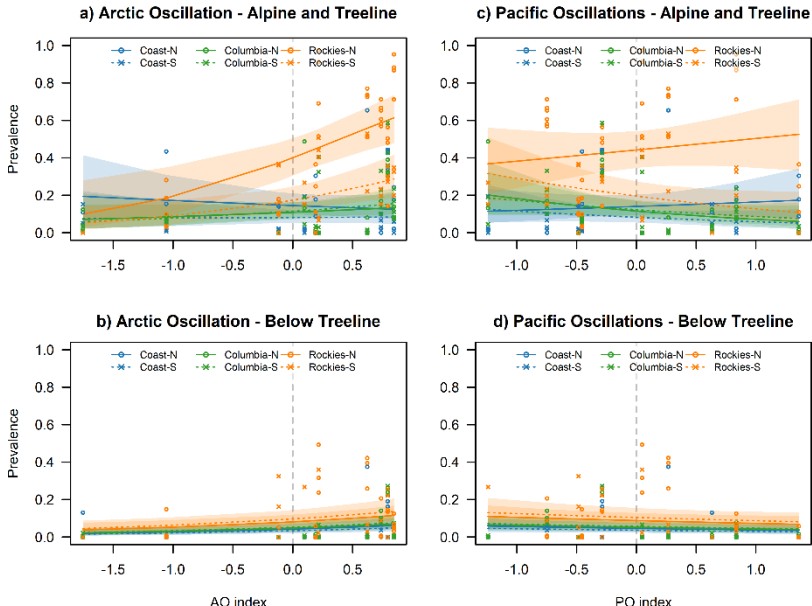

**Figure 8: Calculated prevalence for Deep persistent slab avalanche problems in relation to AO (left column) and Pacific-centered oscillations (right column) for the alpine/treeline (top row) and below treeline (bottom row) over the range of observed index values. Lines indicate the mean estimates and shaded areas represent that 95% confidence intervals. Individual points represent observed prevalence values.**

We also observe an increase in the prevalence of *Wind slab avalanche problems* in the alpine/treeline model, predominantly in the Rockies-North and Columbia-South regions due to significant interactions (Figure 4 and 7). Below treeline, we observe a minimal and only marginally significant positive increase, but wind slabs are rarely forecasted at lower elevations anyway. The observed increase in *Wind slab avalanche problems* is possibly related to the more intense westerly flows caused by the stronger meridional pressure gradient during the positive phase of AO (Fleming et al., 2006; Moore et al., 2009). The more intense westerly and therefore onshore flow might also contribute to the observed increase in the prevalence of *Storm slab avalanche problems* in the Coast-North region (Figure 4 and 5).

The increase in *Deep persistent slab*, *Wind slab*, and *Storm slab avalanche problems* in different parts of the study area during the positive AO phase is potentially compensated by a concurrent decrease in the prevalence of *Dry loose avalanche problems* (Figure 4). While we observe a uniform decrease across the entire study area at all elevation bands, additional interactions in the alpine/treeline model indicate that the effect is weaker in the Columbia Mountains and particularly strong in the Rockies-North region. This pattern is consistent with the stronger impact of AO observed in the Rockies-North region in general.

Another regional response pattern associated with AO is a consistent significant increase in the prevalence of *Wet loose dry avalanches* in the northern parts of the study area (Figure 4). This result is in line with the higher springtime temperatures in north-western British Columbia described by Fleming et al. (2006), but most of the regions included in our study are substantially further south than their study area. The lack of a similar pattern in the below treeline model might be due to the

smaller size of the dataset or the fact that higher elevations are more sensitive to temperature changes in the spring when the lower elevations experience above freezing temperatures anyway. It is worth noting that the AO analysis of Thumlert et al. (2014) did neither find a relationships with avalanche activity overall or dry and wet avalanches separate.

Different from the effect of PO, our analysis did not reveal a significant relationship between the prevalence of days with *No avalanche problems* and AO. Furthermore, the prevalence of *Cornice avalanche problems* was not affected by either oscillation.

## 5 Discussion

The large-scale patterns emerging from our analysis agree reasonably well with the existing understanding of the effect of the Pacific-centered oscillations on the winter weather in BC and the Pacific Northwest. The generally higher temperatures 510 and lower precipitation during the positive phase winters are associated with a decrease in the prevalence of *Storm slab* and *Wind slab avalanche problems* and an increase of days with *No avalanche problems* below treeline. We also see a small increase in the prevalence of *Wet slab avalanche problems* in the alpine/treeline models of the southern part of the study area and potentially some unique local responses in the southeast corner of BC.

The effects of AO are generally more pronounced in the eastern part of the study area and particularly strong in the Rockies-515 North region. The most prominent effects are a strong positive relationship with the prevalence of *Deep persistent* and *Wind slab avalanche problems*, a weak but significant positive relationship with the prevalence of *Wet slab avalanche problems*, and a negative relationship with the prevalence of *Dry loose avalanche problems*. While the stronger/weaker westerly flow during the positive/negative phase of the AO can potentially explain the changes in the prevalence of the *Wind slab* and *Dry loose avalanche problems*, the mechanism underlying the prevalence change in the *Deep persistent slab avalanche problems* 520 is more unclear.

Overall, our analysis revealed strong relationships between the oscillations and avalanche problems types that link directly to meteorological variables like snowfall (*Storm slab avalanche problem, Dry loose avalanche problem*), temperature (e.g., *Wet slab avalanche problem, Wet loose avalanche problem*) or wind (*Wind slab avalanche problem*). Relating the prevalence values of these avalanche problem types to the known characteristics of the PO and AO is relatively 525 straightforward. However, we found much fewer significant relationships with avalanche problem type that are the result of sequences of weather events (*Persistent* and *Deep persistent slab avalanche problems*). Particularly interesting is that no significant effects were identified for the prevalence of *Persistent slab avalanche problems*. Several explanations for this observation are possible: a) the weather sequences required for the development of *Persistent slab avalanche problems* are not related to the atmosphere-ocean oscillations included in this study; b) the seasonally averaged oscillation indices do not 530 describe the oscillations in a way that allows the relationships to emerge; or c) our monotonic analysis approach is unable to detect the more complicated relationship.

Even though the patterns that emerged from our analyses seem to provide a meaningful perspective on the effect of atmospheric oscillations and avalanche hazard in western Canada, there are several important limitations to consider. The most important limitation of our study is the relative shortness of our observation time series. Even though the oscillation indices cover a substantial part of their historic range within our study period, avalanche hazard assessment time series of ten (Avalanche Canada) and eight winters (Parks Canada) are generally too short for climatological studies. While the observed patterns seem to match well with the known effects of the included oscillations, they primarily reflect the nature of the events experienced during the study period, and the generalizability of the results is currently uncertain. In addition, the associations between PO and avalanche hazard presented in this study represented the combined effect of the Pacific-centered atmosphere-ocean oscillations. Isolating the effect of ENSO, PDO and PNA would require a considerably longer time series of avalanche hazard assessments, which are currently not available. Nevertheless, we believe that our current results clearly highlight the potential of our analysis approach for improving our understanding.

It is also important to recognize that Rockies-North is the only region that includes hazard assessment from Parks Canada and Alberta Parks, whereas the assessments in all other regions are produced exclusively by Avalanche Canada. Hence, some of the observed differences in the Rockies-North region may originate from differences between agencies. While avalanche hazard assessment datasets are susceptible to changes in operational practices similar to avalanche observations time series, our knowledge of the change in forecasting practices at Avalanche Canada in 2012 allowed us to explicitly account for it by including an extra parameter in the *Storm slab* and *Wind slab avalanche problem* models. However, it is not possible to completely eliminate the impact of this change on the results, and the patterns for these two avalanche problem types should therefore be interpreted with some caution.

As mentioned in the discussion of the relationship between the oscillations and the prevalence of *Persistent slab avalanche problems*, the logit link of the beta regression can only capture monotonic relationships between the prevalence of an avalanche problem type and the AO and PO indices. Hence, the somewhat surprising lack of an effect may be an artefact of our analysis method. Possible approaches for examining the relationship between the prevalence of *Persistent slab avalanche problems* and the AO and PO indices in more detail include a) converting the numeric oscillation indices into ordinal variables (negative, neutral, positive) and interacting these with the region variable, b) adding quadratic terms to the regression analysis (see, e.g., Fleming and Dahlke, 2014a) or c) using generalized additive models (Wood, 2017). However, at this point, our dataset is far too small for any of these approaches.

## 6 Conclusion

This study presents a new approach for providing insight into the relationship between atmosphere-ocean oscillations and the seasonal character of avalanche hazard. Instead of using avalanche activity records from safety programs along transportation corridors as done by previous studies (McClung, 2013; Thumlert et al., 2014), we used avalanche hazard assessments published in public avalanche bulletins from Avalanche Canada (2010-2019), Parks Canada (2012-2019) and

Alberta Parks (2012-2019) to examine this relationship in western Canada. After summarizing the seasonal nature of avalanche hazard for each forecast area with a set of ten avalanche problem prevalence values, we applied a series of beta mixed-effects regression models to explore the effect of the atmosphere-ocean oscillations known to affect winter weather in western Canada. These included the Pacific-centered and tightly linked El Niño-Southern Oscillation, Pacific Decadal Oscillation and the Pacific North America Teleconnection Pattern as well as the more independent Arctic Oscillation.

We believe that our approach complements and expands previous research in this area in several ways. First, the use of structured avalanche hazard assessments from public avalanche bulletins overcomes some of the inherent limitations of avalanche observations. The consistency and substantial spatial coverage of avalanche bulletins in western Canada offers a more comprehensive perspective of the response of avalanche hazard to atmosphere-ocean oscillations than the focus on point locations of previous studies. Since the meteorological signal of the oscillations, particularly the precipitation signal, have been shown to vary considerably in space (e.g., Fleming and Whitfield, 2010; Wise, 2010; Jin et al., 2006), the increased spatial coverage is critical for beginning to understand the regional differences in the avalanche hazard response. Despite the challenges in the application of the CMAH in public avalanche forecasts recently highlighted by Statham et al. (2018b) or Clark (2019), we believe that the judgment process of avalanche forecasters adds considerable value to the insight gained from such climate analyses. Second, the focus on avalanche problems links the analysis directly to established types of avalanche risk management concerns, which makes the results more relevant and practical for practitioners. While avalanche forecasters might differ in their detailed characterization of avalanche problems and the level of the resulting avalanche danger rating, the identification of the problems by itself is likely less susceptible to forecaster bias, even though differences between agencies may still exist. The third and final advantage of the present study over previous research is the multivariate, model-based approach of the analysis. While the study period was too short to examine the responses to the different Pacific-centered oscillations independently, the regression approach has the potential to properly separate the effects from multiple oscillations, which is not possible with the correlation measures used in previous studies.

With the predictability of the most important atmosphere-ocean oscillations continuously improving, this study contributes towards the knowledge necessary for taking advantage of routine atmosphere-ocean oscillation predictions to create informative seasonal avalanche forecasts for western Canada in a relatively simple way. However, more work is required to properly capitalize on this opportunity. To facilitate future research in this area, we encourage avalanche safety agencies to further strengthen and standardize the use of the CMAH in avalanche hazard assessments. While it is unreasonable to expect avalanche hazard assessment and mitigation practices not to change in the future, properly documenting such changes is critical for allowing long-term studies to account for them in a meaningful way. Future research should also include suitable weather and snowpack observations to provide more insight into the mechanisms responsible for the changed hazard conditions. Since climate models are getting to the point where they can reliably forecast atmosphere-ocean oscillations (e.g., L'Heureux et al., 2017; Fuentes-Franco et al., 2016), this research direction might eventually also contribute to a better understanding of the effect of climate change on avalanche hazard in western Canada and beyond.

**Data and code availability**

The data, code and output for our analysis and the data and code for the figures and tables included in this manuscript are available at https://osf.io/7xsfj/.

**Author contributions**

PH and BS conceptualized the study. PH and BS extracted and processed the data. PH and PM analyzed the data, and PH prepared the manuscript with contributions from all co-authors.

**Competing interests**

The authors declare that they have not competing interest.

**Acknowledgments**

We would like to thank Avalanche Canada, Parks Canada and Alberta Parks for providing access to their avalanche bulletin data. We also thank Simon Horton and three anonymous reviewers for their valuable comments on earlier versions of this manuscript. Thanks also go to Jürg Schweizer for serving as the editor for this manuscript.

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

**Appendix: Beta Mixed-Effects Model Formulation**

Let **X** denote the predictor matrix of $p + 1$ columns ($p$ predictors plus intercept), and $Y_{ij}$ denote a single response observation $i$ (e.g., seasonal prevalence of storm slab avalanche problem, wind slab avalanche problem, persistent slab avalanche problem, etc.) within a season $j$ (season as grouping/clustering variable). Note that each $Y_{ij}$ is bounded within a $[0,1]$ interval (prevalence values). To model the relationship between the predictors and the bounded response with clustered observations we use a beta mixed-effects regression model, which belongs to the class of *generalized linear mixed-effects models* (GLMM). The model can be formulated as follows:

$$g(\mu_{ij}) = \mathbf{x}'_{ij}\boldsymbol{\beta} + b_{0j},$$

with $\mu_{ij}$ as the conditional mean $E(Y_{ij}|\mathbf{x}_{ij})$ of the beta distribution, and $g(\cdot)$ as the link function. In our analyses we use a logit link, therefore $g(\mu_{ij}) = \log\left(\mu_{ij}/(1 - \mu_{ij})\right)$. Further, $\mathbf{x}_{ij}$ is a predictor vector of length $p + 1$. The vector $\boldsymbol{\beta} = (\beta_0, \beta_1, \ldots, \beta_p)'$ are the fixed-effects regression parameters. Finally, $b_{0j}$ is the random effect (random intercept) for season $j$ $b_{0j} \sim N(0, \sigma_{b0}^2)$, where $\sigma_{b0}^2$ is the random effects variance.

We now give a few examples in R formula syntax, representing instances of the general model expression from above:

```
## Persistent slab main effects model (Btl):
Pers ~ AO + PO + Region + (1|Season)
## Persistent slab interaction model (Btl):
Pers ~ AO * Region + PO * Region + (1|Season)
## Persistent slab main effects model (Alp/Tl):
Pers ~ AO + PO + Region + Elevation + (1|Season)
## Persistent slab interaction effects model (Alp/Tl):
Pers ~ AO * Region + PO * Region + Elevation + (1|Season)
```