# Peer review of "Using avalanche problems to examine the effect of large-scale atmosphere-ocean oscillations on avalanche hazard in western Canada"

_The Cryosphere, 2020_

## Referee Comment (RC1) · Anonymous Referee #1 · 3 Jul 2020

I believe the paper is very good on snow avalanche information. The authors are right on the difficulty of researching climate and avalanche data from point locations. Therefore, I like the idea on the use of public bulletins. The CMAH related method on avalanche characteristics has been well studied by the authors in previous work. The violin plots in Fig 2 look good. I like the prevalence methods and approaches with the avalanche types.

Unfortunately, the climate basis for the paper definitely needs a lot of thinking and re-doing, and this is my main comment. Their avalanche data cover 2012-2019, clearly limited for climate research use on what the authors were doing. They stated limita-

tions but it mostly comes at the end and not early in the paper. Given this limitation of a 10 year dataset, I am critical on the use of correlations and other statistics given the sample size for confidence intervals, significance and other things. It would be better throughout to focus on the visualization (as they do in Figure 3 very well). The teleconnection indices such as PNA mostly reflect very broad patterns (probably explain at most about 60 percent of the variance in western Canada), as these specific well-known teleconnections really focus on centres of action of circulation centers at very large scales. Also, the AO impact is really more farther north and more North Atlantic centered. The PDO and to some degree the AO suffers from the autocorrelation issues that can extend for several years (see Deser et al article in J of Climate that debates the clarity of the PDO), and this is quite problematic when connecting results with a 10 year avalanche data set. It might be more conducive to employ synoptic classifications more keen to western Canada, such as work done by Ian McKendry and his group. The McKendry synoptic work also may relate much more to how weather is connected to some of the important snowpack processes. Another suggestion is to perhaps the authors devised their own index from circulation data from gridpoints off the BC coast to reflect more specific aspects of the Pacific subtropical high, low pressure off the coast, etc. On Line 229. I don't like the idea of averaging the teleconnection indices since they have some intercorrelation with one another, plus with the PDO's autocorrelation and seem confusing/artificial in meaning. On Line 39. Arctic is misspelled. The paper is clearly a detailed snow avalanche paper. In the beginning of the paper, it should have a paragraph to appeal more to the broad The Cryosphere readership and why avalanches are significant in Cryosphere studies overall.

---

## Referee Comment (RC2) · Anonymous Referee #2 · 22 Jul 2020

General comments:

This manuscript provides an interesting window into climatological drivers of avalanche hazards of different types and in different regions of western Canada. The study appears to be well-designed and carefully executed overall, and the paper is well-organized. It seems to have potential to be a significant and socially relevant contribution to understanding intersections between climate science and cryospheric science. The authors' closing point is an excellent one: that given the existing, and continuously improving, ability of the climate community to forecast ENSO and other seasonal climate conditions a few months in advance, these climate-avalanche relationships have potential practical advantage for general avalanche season forecasts. The article seems suitable for publication pending minor revisions.

Specific comments:

Some additional referencing for AO impacts in the mountains of western Canada could be helpful. The impacts of Pacific-centered circulation patterns are very well-studied there, but the AO less so. Off the top of my head I can think of four additional examples that could be useful to cite here for additional support. Vincent et al. (2015) mapped out statistically significant temperature and precipitation teleconnections for much of Alberta and British Columbia to the NAO, which is closely related to the AO. Other examples I'm aware of in the region are framed in terms of hydroclimatic teleconnections to the AO specifically: Gobena et al. (2013), Fleming and Dahlke (2014), and Fleming et al. (2016).

While the small sample size (n=10) in time is of concern when studying associations between interannual climate variability and avalanche impacts, these concerns are mitigated by the use of statistical significance tests as an objective basis for ascertaining the presence of associations, as these of course account for sample size. This was done in the study, but the catch here is that the effective sample size might be reduced by serially correlated observations. If such a lack of independence exists, it can be handled in the statistical modeling, but I didn't see any mention of it in the paper. This procedural detail ought to be addressed.

The wording of lines 365-373 require some fine-tuning. The description of PO impacts provided in this passage is accurate for southern British Columbia, but there is a spatial dipole between ENSO/PDO teleconnections in southern BC and the Pacific Northwest vs. those in northwestern British Columbia and southern Alaska. See the article by Fleming and Whitfield (2010) already cited in the article.

References:

Fleming SW, Dahlke HE. 2014. Parabolic northern-hemisphere river flow teleconnections to El Niño-Southern Oscillation and the Arctic Oscillation. Environmental Research Letters, 9, 104007.

Fleming SW et al. 2016. Seasonal flows of international British Columbia-Alaska rivers: the nonlinear influence of ocean-atmosphere circulation patterns. Advances in Water Resources, 87, 42-55.

Gobena AK et al. 2013. The role of large-scale climate modes in regional streamflow variability and implications for water supply forecasting: a case study of the Canadian Columbia River basin. Atmosphere-Ocean, 51, 380-391.

Vincent LA et al. 2015. Observed trends in Canada's climate and influence of low-frequency variability modes. Journal of Climate, 28, 4545-4560.

---

## Author Comment (AC1) · 16 Sep 2020

Please see attached PDF for our detailed responses to the comments from all referees.

Please also note the supplement to this comment:
https://tc.copernicus.org/preprints/tc-2020-146/tc-2020-146-AC1-supplement.pdf

---

## Author Comment (AC2) · 16 Sep 2020

**Responses to Referee #1**

**GENERAL COMMENT**

We thank Referee #1 for their constructive review and helpful comments. We appreciate the encouraging comments about the use of public avalanche bulletin data and the analysis of avalanche problem type information, as well as the suggestions for alternate analysis approaches. Please see below for our detailed responses to specific comments and suggestions from Referee #1. Additions to the manuscript are included in our responses in quotes. In the revised manuscript, the edits are marked with the number of the Referee comment.

**RESPONSES TO SPECIFIC COMMENTS**

**1.1 Limited length of study period**

**Referee Comment:**

*Their avalanche data cover 2012-2019, clearly limited for climate research use on what the authors were doing. They stated limitations but it mostly comes at the end and not early in the paper.*

**Author Response:**

We completely agree with the Referee's concern about the relatively short study period. To account for this issue, we employed a 'conservative' analysis approach, and we were careful not to overinterpret our results. To address the Referee's concern, we included for following additional sentence at the beginning of section 3.3 (Statistical analysis) to alert the reader of this limitation of our study right up front.

*"While a 10-year dataset is relatively short for a climatological study, our analysis approach aims to maximize the value of the available data to provide meaningful insight into the relationship between the combined Pacific-centered atmospheric-ocean oscillations and the AO and the nature of avalanche hazard in western Canada at the regional scale."*

**1.2. Concerns about statistical analysis**

**Referee Comment:**

*Given this limitation of a 10 year dataset, I am critical on the use of correlations and other statistics given the sample size for confidence intervals, significance and other things. It would be better throughout to focus on the visualization (as they do in Figure 3 very well).*

**Author Response:**

As mentioned in our response to comment 1.1, we believe that our chosen 'conservative' statistical approach is appropriate, and the resulting insights are meaningful within the existing constraints. To maximize the value of our short data set and avoid over-interpretation we focused on regional patterns, considered both the statistical significance and the magnitude of the observed associations in our discussion, and related the results to physical processes responsible for the observed associations.

The correlation analyses included in our study primarily serve to illustrate the nature of the available dataset and justify our choices for the analysis. In other words, the correlation coefficients for the atmospheric oscillations presented in Section 3.2 (Information on atmosphere-ocean oscillations) aim to show the similarities of the oscillation indices during our study period (as further highlighter in Fig. 3) and are not intended to characterize the general relationship among the oscillations. Hence, we believe that our approach was justified, and we did not make any modification to the manuscript.

As pointed out by Referee #2, the regression analysis approach and subsequence post-hoc tests employed in our study take sample size into account. Hence, the results presented are statistically significant given the limitations of the dataset.

**1.3 Relevance of PNA and AO**

**Referee Comment:**

*The teleconnection indices such as PNA mostly reflect very broad patterns (probably explain at most about 60 percent of the variance in western Canada), as these specific well known teleconnections really focus on centres of action of circulation centers at very large scales. Also, the AO impact is really more farther north and more North Atlantic centered.*

**Author Response:**

We agree with the reviewer that the atmosphere-ocean oscillations reflect broad patterns and general trends. Our selection in oscillations in the present study was based on a) what was examined in existing snow and avalanche climatology studies, and b) oscillations whose effect on the winter weather in Western Canada has been studied. Please see our response to Referee comments 2.1 and 2.3 for our improved descriptions of the AO and PO

**1.4 Autocorrelation issues**

**Referee Comment:**

*The PDO and to some degree the AO suffers from the autocorrelation issues that can extend for several years (see Deser et al article in J of Climate that debates the clarity of the PDO), and this is quite problematic when connecting results with a 10 year avalanche data set.*

**Author Response:**

We appreciate that the Referee has made us aware of additional relevant PDO literature. In response to this comment, we included Newman et al (2016) as an additional reference in our general discussion of the PDO.

We understand that the monthly indices of atmospheric oscillations like the PDO or AO exhibit considerable autocorrelations. For example, Newman et al. (2016) points out that the year-to-year PDO autocorrelation is over 0.45 in later winter and spring. However, since the seasonal snowpack largely melts out every summer, and the snowpack structures relevant for avalanches emerge each winter independently from the previous winter, it is not necessary to use an autoregressive model approach for the present analysis. We added the following new paragraph in Section 3.3 (Statistical analysis) to explain this situation to the reader.

*"It is well known that the indices of atmospheric oscillations like the PDO or AO exhibit considerable autocorrelations. Newman et al. (2016), for example, points out that the year-to-year PDO correlation is over 0.45 in late winter and spring. However, since the seasonal snowpack in Western Canada largely melts out every summer, and the snowpack structures relevant for avalanches emerge each winter independently of the previous winter, it is not necessary to use an autoregressive model approach for the present analysis."*

Newman, M., Alexander, M. A., Ault, T. R., Cobb, K. M., Deser, C., Di Lorenzo, E., Mantua, N. J., Miller, A. J., Minobe, S., Nakamura, H., Schneider, N., Vimont, D. J., Phillips, A. S., Scott, J. D., and Smith, C. A.: The Pacific Decadal Oscillation, Revisited, J. Climate, 29, 4399-4427, 10.1175/jcli-d-15-0508.1, 2016.

**1.5 Averaging of teleconnection indices**

**Referee Comment:**

*On Line 229. I don't like the idea of averaging the teleconnection indices since they have some intercorrelation with one another, plus with the PDO's autocorrelation and seem confusing/artificial in meaning.*

**Author Response:**

Due to the high correlation of the Pacific-centered oscillations within our study period, a regression analysis that includes each of the oscillations (ENSO, PDO, PNA) as separate parameters would not be able to properly isolate their effects. Furthermore, we would be pushing what is reasonable given our relatively short study period. Since there are many similarities in the effects of the Pacific-centered oscillations on the winter weather in Western Canada, averaging their indices for the present analysis seems reasonable. In our opinion, the intercorrelations mentioned by the Referee, the shared correlations to some common underlying process, seems to further justify the averaging of the indices. However, to address this concern, we added the following two new sentences in the limitation paragraph of the new discussion section:

*"The associations between PO and avalanche hazard presented in this study represented the combined effect of the Pacific-centered atmosphere-ocean oscillations. Isolating the effect of ENSO, PDO and PNA would require a considerably longer time series of avalanche hazard assessments, which are currently not available."*

Also see our response to Referee comment 1.4 for our discussion of the concerns around autocorrelation issues.

**1.6 Synoptic classification**

**Referee Comment:**

*It might be more conducive to employ synoptic classifications more keen to western Canada, such as work done by Ian McKendry and his group. The McKendry synoptic work also may relate much more to how weather is connected to some of the important snowpack processes. Another suggestion is to perhaps the authors devised their own index from circulation data from grid points off the BC coast to reflect more specific aspects of the Pacific subtropical high, low pressure off the coast, etc.*

**Author Response:**

We appreciate the suggestions of alternative analysis approaches for our dataset. While we understand the potential benefits of the suggested approaches, the objective of this study was to examine the relationship between avalanche hazard in western Canada and well-established indices of the most dominant atmosphere-ocean oscillations affecting the regional weather conditions. This choice was made to build on existing research and to explore the possibilities for producing seasonal avalanche predictions from existing seasonal forecasts of climate conditions (as highlighted by Referee #2). We will keep the Referee's suggestions in mind for future research.

**1.7 Introduction to avalanche topic for The Cryosphere readership.**

**Referee Comment:**

*The paper is clearly a detailed snow avalanche paper. In the beginning of the paper, it should have a paragraph to appeal more to the broad The Cryosphere readership and why avalanches are significant in Cryosphere studies overall.*

**Author Response:**

Avalanche safety research seems a well-established topic in TC. However, to better introduce the topic to the TC readership, we added the following sentences at the beginning of the introduction describing the societal impact of avalanche hazard.

*"Snow avalanches are an inherent natural hazard in mountainous regions that receive substantial amounts of seasonal snow. In these regions, snow avalanches can threaten communities, transportation corridors, critical infrastructure (e.g., hydroelectric dams, transmission and communication lines, pipelines) and resource extraction operations. In Western countries, most people killed in avalanches are recreationists pursuing winter mountain activities such as backcountry skiing, mountain snowmobile riding and out-of-bounds skiing."*

**1.8 Misspellings**

**Referee Comment:**

*On Line 39. Arctic is misspelled.*

**Author Response:**

Thank you for pointing out this typo. We have fixed the spelling.

**Responses to Referee #2**

**GENERAL COMMENT**

We thank Referee #2 for their constructive review and helpful comments. We appreciate the encouraging comments about the quality of our study and its contribution to the scientific literature. Please see below for our detailed responses to specific comments and suggestions from Referee #2. Additions to the manuscript are included in our responses in quotes. In the revised manuscript, the edits are marked with the number of the Referee comment.

**RESPONSES TO SPECIFIC COMMENTS**

**2.1 Addition information on AO impacts in mountain of Western Canada**

**Referee Comment:**

*Some additional referencing for AO impacts in the mountains of western Canada could be helpful. The impacts of Pacific-centered circulation patterns are very well-studied there, but the AO less so. Off the top of my head I can think of four additional examples that could be useful to cite here for additional support. Vincent et al. (2015) mapped out statistically significant temperature and precipitation teleconnections for much of Alberta and British Columbia to the NAO, which is closely related to the AO. Other examples I'm aware of in the region are framed in terms of hydroclimatic teleconnections to the AO specifically: Gobena et al. (2013), Fleming and Dahlke (2014), and Fleming et al. (2016).*

**Author Response:**

We appreciate Referee #2 highlighting the additional relevant literature to us. After reviewing the suggested papers, we included the following additional sentences in our initial description of the AO.

*"Gobena et al. (2013), who studied the effect of AO on stream flows in the Columbia River Basin of Southeastern BC, only identified effects during negative AO anomalies with cooler than average temperatures during December, January and March, and below-average precipitation during winter and spring. Vincent et al. (2015), on the other hand, noted a positive association of winter temperatures in Northern BC with the North Atlantic Oscillation, a close relative to the AO (Fleming and Dahlke, 2014). They did not find a significant signal in winter precipitation."*

While the other papers do discuss the effect of AO on weather in British Columbia, their descriptions are more focused on summer patterns and seem therefore less relevant.

After reading about the non-monotonic response patterns in the suggested papers, we also included references to them in our discussion of the non-existing relationship between PO and the prevalence of persistent slab avalanche problems. References were included in both Sections 4.1 (Response to Pacific-centered oscillations) and the limitation paragraph of the new discussion section.

Fleming, S. W., and Dahlke, H. E.: Modulation of linear and nonlinear hydroclimatic dynamics by mountain glaciers in Canada and Norway: Results from information-theoretic polynomial selection, Can. Water Resour. J., 39, 324-341, 10.1080/07011784.2014.942164, 2014.

Gobena, A. K., Weber, F. A., and Fleming, S. W.: The Role of Large-Scale Climate Modes in Regional Streamflow Variability and Implications for Water Supply Forecasting: A Case Study of the Canadian Columbia River Basin, Atmos. Ocean, 51, 380-391, 10.1080/07055900.2012.759899, 2013.

Vincent, L. A., Zhang, X., Brown, R. D., Feng, Y., Mekis, E., Milewska, E. J., Wan, H., and Wang, X. L.: Observed Trends in Canada's Climate and Influence of Low-Frequency Variability Modes, J. Climate, 28, 4545-4560, 10.1175/jcli-d-14-00697.1, 2015.

**2.2. Concerns about serially correlated observations**

**Referee Comment:**

*While the small sample size (n=10) in time is of concern when studying associations between interannual climate variability and avalanche impacts, these concerns are mitigated by the use of statistical significance tests as an objective basis for ascertaining the presence of associations, as these of course account for sample size. This was done in the study, but the catch here is that the effective sample size might be reduced by serially correlated observations. If such a lack of independence exists, it can be handled in the statistical modeling, but I didn't see any mention of it in the paper. This procedural detail ought to be addressed.*

**Author Response:**

*This concern was also raised by Referee #1. Please see Referee Comment 1.4 for our response.*

**2.3 Description of PO impacts**

**Referee Comment:**

*The wording of lines 365-373 require some fine-tuning. The description of PO impacts provided in this passage is accurate for southern British Columbia, but there is a spatial dipole between ENSO/PDO teleconnections in southern BC and the Pacific Northwest vs. those in northwestern British Columbia and southern Alaska. See the article by Fleming and Whitfield (2010) already cited in the article.*

**Author Response:**

After reviewing the article of Fleming and Whitfield (2010) again, we expanded our description of the impact of PO on the weather in British Columbia both in Section 2.1 (Atmosphere-ocean oscillation affecting weather in western Canada) and Section 4.1 (Response to Pacific-centered oscillation).

We added the following sentence to Section 2.1:

*"Fleming and Whitfield (2010) highlight that the positive temperature signal of El Niño is weaker in northern BC, and while El Niño tends to bring drier conditions to the southern part of BC, it produces wetter conditions along the northern coast."*

The edited version of the relevant paragraph in Section 4.1 now reads (additional underlined):

*"These observations are consistent with the existing understanding of the effect of PO on the winter weather in the southern parts of BC and the Pacific Northwest as the warmer temperatures experienced during the positive phase (Shabbar and Khandekar, 1996; Shabbar and Bonsal, 2004; Bonsal et al., 2001) generally result in a shallower and less hazardous snowpack at lower elevations. The observed pattern is also consistent with the results of Lute and Abatzoglou (2014), who showed that La Niña winters in the Pacific Northwest are generally associated with above normal snow water equivalents that result from both more snowfall days and more extreme snow fall events compared to El Niño winters, and the*

*studies of Brown and Goodison (1996) and Moore and McKendry (1996) who showed that the positive phases of both ENSO and PNA are associated with reduced snow cover western Canada. Hence, our prevalence values for alpine/treeline Storm slab avalanche problems exhibit the expected negative association with PO at higher elevations (Figure 4 and 5). Consistent with the previous research, our regression analysis indicates a homogeneous effect of PO across the study area (i.e., no significant interaction effect), but the magnitude of the estimated difference over the observed PO index is most pronounced in the Rocky Mountains. While Fleming and Whitfield (2010) point out that the northern coast of BC and Alaska exhibits an inverse response pattern for precipitation with the warm ENSO phase bringing wetter winter and spring conditions, this deviation would only affect the Coast North region of our study area."*

**Responses to Editor**

**RESPONSES TO SPECIFIC COMMENTS**

**3.1 Conclusions too long**

**Editor Comment:**

*The Conclusions are somewhat lengthy and I recommend you consider, during the revision process, to move some paragraphs into a Discussion section.*

**Author Response:**

In response to this comment we split the original results and discussion section into two separate sections. The revised results section now focuses on the presentation of the results of the beta regression models. The new discussion section includes a concise summary of the results that contains additional big-picture interpretations (previously included in conclusion section) and a detailed analysis of the limitations of the study.

---

## Referee Report (RR1)

Thanks you very much for asking me to review the paper. Just to make things clear, I was not involved in the first round of public discussion, but could see the referee comments in the open discussion.

The paper investigates the fluctuations of avalanche activity in Western Canada and its links to large-scale climate patterns. Main novelty of the article is to ground on avalanche problems series that result from avalanche forecasting bulletins. This is an excellent idea that has huge potential for many studies in the snow avalanche field. Also valuable is the chosen statistical methodology, namely a model-based assessment using a generalized linear model adapted to the data structure. Such an approach, even though it is not completely new, remains little used in the snow and avalanche field. Formally, the paper is written in an excellent English, probably far better than mine, so I won't comment any further language issues.

By contrast, the findings of the paper may be seen as a bit weak. We indeed already know that avalanche activity is weakly related to some atmospheric circulation patterns (see, e.g. Birkeland et al., 2001; Keylock 2005; Garcia Selles et al., 2010, Oeller et al., 2015) and even than this is true in Canada (McClung, 2013). The paper somewhat refine and reinforce this statement (different types of avalanches, different atmospheric indexes), but, from "outside the box", not that much, and it is not fully clear to which extent the results obtained may therefore be useful for a broad readership. This discussible added value to the state of the art regarding main drivers of avalanche activity is first due to the short period analyzed : you cannot expect to infer very significant patterns at a decadal time scale with ten years of data. Second there is also a more fundamental issue regarding how meaningful/useful it is to search for some linkages between avalanche activity and synoptic patterns rather than focusing on more direcrt local climate patterns. From a different perspective, the basics of the statistical method used, although clearly adapted to the data structure and amount, is very hard to follow and is not discussed in the paper.

All in all, there are some very interesting and innovative points in what the authors are proposing (avalanche problems as data source and model-based statistical analyses) but in my opinion they are insufficiently put forward in the paper which focus much more on results which do not appear as fully conclusive and/or "useful". I therefore highly recommend publication of the paper but after substantial reworking of the text, so as to more clearly present the overall benefit of the work done.

**Avalanche problems from forecasting bulletins as data source**

This is really the new, great, idea introduced by the paper. This provides regular daily information about the potential occurrence of different types of avalanches (referenced as avalanche problems). As a consequence, it has certainly lots of potential and will arguably be heavily used in other studies due to the recurrent lack of long and homogeneous data series in the field. A really see such an approach as a new way of providing avalanche data series usable for various scientific questions in all areas where regular forecasting bulletins are issued. I however see two issues in what the authors are proposing:

- The authors state at several points the superiority of their data with regard to real avalanche observations because of a higher homogeneity and because of being more informative regarding different types of avalanches. Again, I like the idea and the data, but a more "modest" posture would be preferable. As the authors themselves show, homogeneity is also an issue for such data (they have to distinguish data before/after 2012 in their analysis). Also, it exists high quality series of observed avalanches likely to provide insights of past changes, even of different avalanche types and over longer time periods (see, e.g. Eckert et al., 2013 for changes in avalanche flow regimes or Naaim et al., 2016 for changes in dry/dense flow type from observed data). Other types of data such as historical archives and indirect proxy data (e.g. Giacona et al., 2017) also provide interesting insights (e.g. Ballesteros-Canovas et al., 2018). Eventually, to which extent avalanche problems series reflect real avalanche

activity remains somewhat unclear and is clearly a source of bias. Hence, rather than a better data source, it is one more, with different strength / weaknesses, which is already a lot, but could/should be discussed in a more comprehensive and fair way.

- The author process a ~10 year long data series. The statistical method chosen is consistent with this time frame (see below), but, anyhow, it is clearly short for investigating long term changes, even at the "decadal" time scale corresponding to synoptic patterns. So wouldn't it be possible to generate longer avalanche problems series further back in the past even if operational forecasting did not exist at that time? I assume longer snow and weather records exist, and it could be possible to ask forecaster to issue some past bulletins on this basis, to use some machine learning techniques or even to combine both approaches to generate past series of daily avalanche problems. This would clearly make available information more insightful in terms of long-term changes.

*Linkages to synoptic patterns*

The authors propose a detailed analysis of the linkages between the different avalanche problems and various (in fact many) large scale atmospheric patterns. The study is done in a very rigorous way and results are deeply analysed, I have nothing to say about it except that : 1) it is long due to the high number of atmospheric indexes used and 2) it is not fully conclusive. The latter lack of strong result is not due to an inappropriate statistical methodology, but arguably to i) the short time frame covered ii) the possibly always weak link between large scale atmospheric patterns and local avalanche activity.

From the perspective of the last point my question is more "philosophical" about the interest of such an analysis. I know this is a topic that has some place in the snow avalanche field (e.g. Keylock 2005 and other references above) but I am always a bit unsure about the real added value in terms of short and long term forecasting. I mean, to interpret the results, it is always necessary to use local climate conditions as an intermediate. For example, l. 500 the authors interpret the positive linkage between wind slab avalanches and arctic oscillation by intense westerly flows. As a consequence, why is it so useful to highlight a weak link to AO? Is AO in the future easier to predict that regional snow and weather conditions which are clearly much more direct predictors of local avalanche activity? I am far from sure… Hence, why the focus of the study is on the link with synoptic patterns given the 10 year data series at hand seems to me unclear. This is all the more true that the authors use avalanche problems that do not necessarily reflect real activity. I would have expected first a detailed analysis of how avalanche problems series relate to real avalanche activity and local snow and weather conditions…

All in all, as I do not ask to change everything, I would suggest to reduce significantly the number of considered indexes, sticking on the most significant ones that may really bring "something" to our broad knowledge of the links between synoptic patterns and avalanche activity. At the same time, adding "intermediate" snow and weather data, as well as potentially local avalanche activity series (if these exist) would help really understand the results. Namely it should then be able to answer why there is a relation, strong or weak, between the avalanche activity which is observed locally and the synoptic pattern. Maybe there is no link because avalanche problems do not reflect real activity, or because local snow conditions are very poorly related to the synoptic pattern… In any case I would suggest that the authors elaborate on this points. This may be of broader interest for the readership than knowing if, on a 10 year time frame, AO is a bit better correlated to some avalanche problems in Canada than ENSO, for example.

*Statistical modelling*

Authors' statistical strategy relies on the use of generalised linear models with a Beta likelihood adapted to model standardized proportions. This offers a suitable framework to investigate whether or not some effects (such as a synoptic pattern) is significant or not. I fully agree that this is a much better strategy given the 10 year data available than relaying only on correlation analyses, tests, p-values, etc. As stated earlier, such an approach is

not fully new in the snow avalanche field (see., e.g. Eckert et al., 2010; Lavigne et al., 2015; 2017), but remains insufficiently used. However, I see two issues:

- The model is not presented in a formal, mathematical way. This makes very difficult to follow exactly what is done and especially how the stratification is built (what is grouped or not, etc.) and what are the different fixed and random effects that are considered. As this is quite frustrating, and should be changed,. At least the authors could include a devoted Appendix if they do not want any equations in the core of their text (but we are in a scientific journal, after all…)
- Model based statistics is the right way of processing small data samples, that's true (see e.g. Diggle 2007). You are gaining inferential power thanks to the model structure. In other words, effects/relations become more easily significant because of the modelling assumptions than with purely non-parametric data-based approaches. By contrast, you have to pay a price (nothing is granted for nothing) and that's the modelling assumptions. So this should be investigated/discussed somewhere, which is currently barely the case. Among potential issues: what about standard model fit to data diagnoses? And scores to evaluate whether or not other model structures than the one chosen would be more suitable? Eventually, I am wondering if the data content could not be more informatively used. As far as I understand, the 8 proportions are processed as independent quantities whether arguably some combinations are more likely than others. Could this be taken into account into the modelling? Same for space, could some kind of distance between regions be included (I assume close regions are more likely to behave in a similar way), etc.

**Reference choices**

The list is impressive. However it is strongly biased towards Canadian studies (or at least studies carried out in North America). This precludes discussing the approach and results in a broad context, and notably to highlight the main strength and weaknesses of what is proposed for a broad readership not especially interested in avalanche regime in western Canada. According to the reference earlier I would suggest the authors to insist more on existing knowledge on avalanche – synoptic patterns relations, avalanche data series and ong term changes, and theire processing with advanced model based statistical techniques.

**Further formal aspects:**

- Abstract seems way too long for an abstract
- Organization is a bit awkward. I assume that sect. 2 content could be easily moved to other sections (introduction, discussion).
- The paper is quite lengthy. Given that in my opinion, the main interest lies within the method (avalanche problems as data source and model-based statistical analyses) rather than within the results regarding linkages to synoptic patterns, it could certainly be significantly shortened without losing the key message, and focusing on the main novelties of broad interest.
- Figures 6-8 look a bit too much like direct outputs of a statistical software (namely R probably). I am wondering if something more "visual" and easy to read could be produced? Others figures are nice.

**Reference**

Ballesteros-Cánovas, J. A., Trappmann, D., Madrigal-González, J., Eckert, N., & Stoffel, M. (2018). Climate warming enhances snow avalanche risk in the Western Himalayas. *Proceedings of the National Academy of Sciences*, *115*(13), 3410-3415.

Birkeland, K. W., Mock, C. J., & Shinker, J. J. (2001). Avalanche extremes and atmospheric circulation patterns. *Annals of Glaciology*, *32*, 135-140.

Diggle, P. J., & Ribeiro, P. J. (2007). Model-based geostatistics (Springer series in statistics).

Eckert, N., Keylock, C. J., Castebrunet, H., Lavigne, A., & Naaim, M. (2013). Temporal trends in avalanche activity in the French Alps and subregions: from occurrences and runout altitudes to unsteady return periods. *Journal of Glaciology*, *59*(213), 93-114.

Eckert, N., Parent, E., Kies, R., & Baya, H. (2010). A spatio-temporal modelling framework for assessing the fluctuations of avalanche occurrence resulting from climate change: application to 60 years of data in the northern French Alps. *Climatic change*, *101*(3-4), 515-553.

García-Sellés, C., Peña, J. C., Martí, G., Oller, P., & Martínez, P. (2010). WeMOI and NAOi influence on major avalanche activity in the Eastern Pyrenees. *Cold Regions Science and Technology*, *64*(2), 137-145.

Giacona, F., Eckert, N., & Martin, B. (2017). A 240-year history of avalanche risk in the Vosges Mountains based on non-conventional (re) sources. *Natural Hazards & Earth System Sciences*, *17*(6).

Keylock, C. J. (2003). The North Atlantic oscillation and snow avalanching in Iceland. *Geophysical Research Letters*, *30*(5).

Lavigne, A., Eckert, N., Bel, L., & Parent, E. (2015). Adding expert contributions to the spatiotemporal modelling of avalanche activity under different climatic influences. *Journal of the Royal Statistical Society: Series C: Applied Statistics*, 651-671.

Lavigne, A., Eckert, N., Bel, L., Deschâtres, M., & Parent, E. (2017). Modelling the spatio-temporal repartition of right-truncated data: an application to avalanche runout altitudes in Hautes-Savoie. *Stochastic Environmental Research and Risk Assessment*, *31*(3), 629-644.

McClung, D. M. (2013). The effects of El Niño and La Niña on snow and avalanche patterns in British Columbia, Canada, and central Chile. *Journal of Glaciology*, *59*(216), 783-792.

Naaim, M., Eckert, N., Giraud, G., Faug, T., Chambon, G., Naaim-Bouvet, F., & Richard, D. (2016). Impact du réchauffement climatique sur l'activité avalancheuse et multiplication des avalanches humides dans les Alpes françaises. *La Houille Blanche*, (6), 12-20.

Oller, P., Muntán, E., García-Sellés, C., Furdada, G., Baeza, C., & Angulo, C. (2015). Characterizing major avalanche episodes in space and time in the twentieth and early twenty-first centuries in the Catalan Pyrenees. *Cold Regions Science and Technology*, *110*, 129-148.

---

## Author Response (AR2)

**Responses to Referee #3**

**GENERAL COMMENT**

We thank Referee #3 for their constructive review and helpful comments. We appreciate the encouraging comments about the potential of including avalanche problem information and the value of the model-based assessment approach we used in our study. Please see below for our detailed responses to specific comments and suggestions from Referee #3. Additions to the manuscript are included in our responses in quotes. In the revised manuscript, the edits are marked with the number of the Referee comment.

**RESPONSES TO SPECIFIC COMMENTS**

**3.1 Value of analysis and focus of paper**

**Referee Comment:**

*All in all, there are some very interesting and innovative points in what the authors are proposing (avalanche problems as data source and model-based statistical analyses) but in my opinion they are insufficiently put forward in the paper which focus much more on results which do not appear as fully conclusive and/or "useful".*

And later

*From the perspective of the last point my question is more "philosophical" about the interest of such an analysis. I know this is a topic that has some place in the snow avalanche field (e.g. Keylock 2005 and other references above) but I am always a bit unsure about the real added value in terms of short and long term forecasting. I mean, to interpret the results, it is always necessary to use local climate conditions as an intermediate. For example, l. 500 the authors interpret the positive linkage between wind slab avalanches and arctic oscillation by intense westerly flows. As a consequence, why is it so useful to highlight a weak link to AO? Is AO in the future easier to predict that regional snow and weather conditions which are clearly much more direct predictors of local avalanche activity? I am far from sure… Hence, why the focus of the study is on the link with synoptic patterns given the 10 year data series at hand seems to me unclear. This is all the more true that the authors use avalanche problems that do not necessarily reflect real activity. I would have expected first a detailed analysis of how avalanche problems series relate to real avalanche activity and local snow and weather conditions…*

*All in all, as I do not ask to change everything, I would suggest to reduce significantly the number of considered indexes, sticking on the most significant ones that may really bring "something" to our broad knowledge of the links between synoptic patterns and avalanche activity. At the same time, adding "intermediate" snow and weather data, as well as potentially local avalanche activity series (if these exist) would help really understand the results. Namely it should then be able to answer why there is a relation, strong or weak, between the avalanche activity which is observed locally and the synoptic pattern. Maybe there is no link because avalanche problems do not reflect real activity, or because local snow conditions are very poorly related to the synoptic pattern… In any case I would suggest that the authors elaborate on this points. This may be of broader interest for the readership than knowing if, on a*

*10 year time frame, AO is a bit better correlated to some avalanche problems in Canada than ENSO, for example.*

**Author Response:**

We appreciate the reviewer's comment and the opportunity to reflect on the value of our study.

In line with the reviewer's perspective, we believe that this manuscript contributes to the literature in two ways: a) by increasing our understanding of the effect of atmosphere-ocean oscillations on the nature of seasonal avalanche hazard in western Canada, and b) by presenting a new analysis approach (inclusion of avalanche problem information, multivariate, model-based analysis approach). However, despite the relatively short study period (which we openly acknowledge in several sections of the manuscript), we disagree with the reviewer that the results are 'weak' and do not warrant a detailed discussion. It is our opinion that the detailed description of the results is crucial for highlighting the validity and value of the analysis approach. Hence, we did not substantially shorten the manuscript or reduce the number of oscillation indices included in the analysis. The nature of avalanche hazard in western Canada shows distinct responses to the Arctic Oscillation and the combined Pacific-centered oscillations. In addition, we believe that combining ENSO, PDO and PNA into a single average index, openly acknowledges that our existing dataset does not allow us to properly isolate the effect of the individual atmosphere-ocean oscillations.

However, the reviewer's reflection on the added value for short- and long-term forecasting, made us realize that the practical motivation for this study (and the broader line of research in general) might not be clear. To address this issue, we expanded the last paragraph of the introduction to better explain our motivation. The revised paragraph reads as follows:

*"The objective of the present study is to complement the existing research on the effect of large-scale atmosphere-ocean oscillations on avalanche hazard in western Canada by taking advantage of the avalanche problem information included in public avalanche bulletins that follow the conceptual model of avalanche hazard (Statham et al., 2018a). This approach links the analysis more closely to backcountry avalanche risk management and overcomes some of the shortcomings of previous studies. Even though linking avalanche hazard conditions to large-scale atmosphere ocean oscillations is unable to provide direct insight for operational, day-to-day avalanche safety decisions, a better understanding of these relationships has the potential to allow the avalanche safety community to take advantage of atmosphere-ocean oscillation predictions that are routinely provided by meteorological services to produce informative seasonal avalanche hazard forecasts. Being able to predict the general nature of seasonal avalanche conditions (e.g., there is a good chance that this winter will be dominated by a deep persistent avalanche problem) would help avalanche professionals and recreationists to develop meaningful risk management expectations for an upcoming season. As pointed out by LaChapelle (1980) and McClung (2002), avalanche forecasting is a dynamic and iterative process that resembles Bayesian updating where having a prior or hypothesis is critical."*

While we previously discussed these ideas in the conclusion section, moving them to the introduction makes them more prominent. Furthermore, it should explain why including additional weather or snowpack observations in in the analysis would go against the objective to find a 'cheap' way to produce seasonal avalanche forecasts, which is distinctly different from producing models for offering short-term

insight for operational avalanche safety operations or examining the effect of climate change on avalanche hazard.

We hope that the expanded introduction addresses the reviewer's concern adequately.

**3.2 Strengths and weaknesses of avalanche problem dataset**

**Referee Comment:**

*The authors state at several points the superiority of their data with regard to real avalanche observations because of a higher homogeneity and because of being more informative regarding different types of avalanches. Again, I like the idea and the data, but a more "modest" posture would be preferable. As the authors themselves show, homogeneity is also an issue for such data (they have to distinguish data before/after 2012 in their analysis). Also, it exists high quality series of observed avalanches likely to provide insights of past changes, even of different avalanche types and over longer time periods (see, e.g. Eckert et al., 2013 for changes in avalanche flow regimes or Naaim et al., 2016 for changes in dry/dense flow type from observed data). Other types of data such as historical archives and indirect proxy data (e.g. Giacona et al., 2017) also provide interesting insights (e.g. Ballesteros-Canovas et al., 2018). Eventually, to which extent avalanche problems series reflect real avalanche activity remains somewhat unclear and is clearly a source of bias. Hence, rather than a better data source, it is one more, with different strength / weaknesses, which is already a lot, but could/should be discussed in a more comprehensive and fair way.*

**Author Response:**

We appreciate this comment from the reviewer, and we agree that each dataset has its own strengths and weaknesses. In response to this comment, we have revised several sections of the manuscript to better reflect that our avalanche problem dataset and analysis simply provide a different perspective that complements the existing studies. See below for the text of the revised sections (marked with a comment "Reviewer comment 3.2" in the revised manuscript.

Introduction
*"Furthermore, changes in avalanche risk mitigation practices along these transportation corridors can add noise to the avalanche activity record that make it more difficult to attribute the observed patterns to changes in winter weather (Bellaire et al., 2016; Sinickas et al., 2016; Jamieson et al., 2017)."*
instead of
*"Furthermore, the observed patterns in avalanche activity are difficult to conclusively attribute to changes in winter weather because the risk from avalanches along transportation corridors is tightly managed, which makes the available avalanche observation time series vulnerable to changes in avalanche control practices (Bellaire et al., 2016; Sinickas et al., 2016; Jamieson et al., 2017)."*

Beginning of Section 2.2
*"One of the challenges for examining the relationship between atmosphere-ocean oscillations and seasonal avalanche hazard is how to describe avalanche hazard in a meaningful way. While existing studies have primarily focused on the frequency of avalanches, the ratio between dry and wet avalanches, or the number of avalanche cycles, Atkins (2004) and Statham et al. (2018a) highlighted that the nature of avalanche hazard, its distribution in the terrain and evolution throughout the season are much more important for avalanche risk management than the frequency of avalanches alone."*

instead of

*"One of the challenges of existing studies examining the effect of atmosphere-ocean oscillations on avalanche hazard is the limited insight into the character of avalanche winters provided by the frequency of avalanches and the ratio between dry and wet avalanches. The nature and severity of avalanche problems, their distribution in the terrain and their evolution throughout the season are much more important for avalanche risk management than the frequency of avalanches alone (Atkins, 2004; Statham et al., 2018a)."*

Discussion section
*"While avalanche hazard assessment datasets are susceptible to changes in operational practices similar to avalanche observations time series, our knowledge of the change in forecasting practices at Avalanche Canada in 2012 allowed us to explicitly account for it by including an extra parameter in the Storm slab and Wind slab avalanche problem models."* (new addition)

Conclusion section
*"We believe that our approach complements and expands previous research in this area in several ways."*
instead of
*"We believe that our approach has several advantages over previous research in this area."*

**3.3. Expanding the avalanche problem dataset**

**Referee Comment:**

*The author process a ~10 year long data series. The statistical method chosen is consistent with this time frame (see below), but, anyhow, it is clearly short for investigating long term changes, even at the "decadal" time scale corresponding to synoptic patterns. So wouldn't it be possible to generate longer avalanche problems series further back in the past even if operational forecasting did not exist at that time? I assume longer snow and weather records exist, and it could be possible to ask forecaster to issue some past bulletins on this basis, to use some machine learning techniques or even to combine both approaches to generate past series of daily avalanche problems.*

**Author Response:**

We completely concur with the reviewer that the limited length of our data set is one of the main limitations of our study, and we clearly state this in discussion section of the paper. While it would be nice to expand the dataset to earlier years, it is currently not possible. Operational avalanche forecasting is an evolutionary process that relies on a continuous integration of a wide range of avalanche safety observation (McClung, 2001). Fully assessing the nature of avalanche hazard of past winters for all of western Canada retrospectively would be an incredible amount of work that would likely not produce a reliable result.

Identifying and characterizing avalanche problems based on snow and weather records using machine learning algorithms is a topic of current research. See, e.g., Horton et al (2020) recently published in NHESS (https://nhess.copernicus.org/articles/20/3551/2020/), or the poster "How close are we to automated avalanche forecasting? Lessons from testing machine learning methods in Norway and Canada" presented by Horton, Müller, Haegeli and Engeset at the 2020 Virtual Snow Science Workshop (https://vssw2020.com/poster-submissions-2/). At this point, we are not in a position to reliably

predicting avalanche problems from snow and weather observations that would contribute to this study in a meaningful way.

Hence, we did not make any modifications to the manuscript in response to this comment.

**3.4 Presentation of model**

**Referee Comment:**

*The model is not presented in a formal, mathematical way. This makes very difficult to follow exactly what is done and especially how the stratification is built (what is grouped or not, etc.) and what are the different fixed and random effects that are considered. As this is quite frustrating, and should be changed. At least the authors could include a devoted Appendix if they do not want any equations in the core of their text (but we are in a scientific journal, after all…)*

**Author Response:**

We now include an Appendix where we give a formal expression of the beta mixed-effects model (in classical Laird-Ware formulation). We also include a few examples in R formula syntax which should help the reader to connect the computational model expression to the mathematical formulation. Please see appendix in revised manuscript for details. We also included a reference to the appendix in the main text of Section 3.3.

Please note that we also provide our full dataset and analysis code for readers interested in the full details of our analysis.

**3.5 Description of model assumptions**

**Referee Comment:**

*Model based statistics is the right way of processing small data samples, that's true (see e.g. Diggle 2007). You are gaining inferential power thanks to the model structure. In other words, effects/relations become more easily significant because of the modelling assumptions than with purely non-parametric data-based approaches. By contrast, you have to pay a price (nothing is granted for nothing) and that's the modelling assumptions. So this should be investigated/discussed somewhere, which is currently barely the case. Among potential issues: what about standard model fit to data diagnoses? And scores to evaluate whether or not other model structures than the one chosen would be more suitable?*

*Eventually, I am wondering if the data content could not be more informatively used. As far as I understand, the 8 proportions are processed as independent quantities whether arguably some combinations are more likely than others. Could this be taken into account into the modelling? Same for space, could some kind of distance between regions be included (I assume close regions are more likely to behave in a similar way), etc.*

**Author Response:**

We appreciate the general approval of our analysis approach. With respect to the model assumptions, we assume that the reviewer's comment refers to regression diagnostics for checking those assumptions. As we are operating within a GLMM framework, simple (normal) residual checks do not work. We therefore used simulated quantile residuals (Dunn & Smyth, 1996) as implemented in the DHARMa package (Hartig, 2020) in R. Visual inspection of the resulting diagnostic plots (e.g., Q-Q-plot

for uniformly distributed residuals) does not suggest any substantial model violations. We added the following new text in the last paragraph of Section 3.3.

*"To assess violations in model assumptions, we simulated quantile residuals (Dunn and Smyth, 1996) as implemented in the DHARMa package (Hartig, 2020). Visual inspection of the resulting diagnostic plots (e.g., Q-Q-plot for uniformly distributed residuals) did not suggest any substantial model violations."*

Regarding "scores to evaluate model structure" we assume that the reviewer is referring to model comparisons. For each avalanche problem we hypothesized two modeling scenarios: 1) a main effects model and 2) an interaction model where both AO and PO interact with the mountain range. Subsequently a likelihood-ratio test was applied to choose between these two hypothesized models. We did this for each response variable (Below treeline and Alpine/Treeline separately). This approach was already described in the manuscript. See highlighted section in updated manuscript with track changes.

We also describe in several places in the manuscript that our beta regression models are only able to capture monotonic relationships between the atmosphere-ocean oscillations and the prevalence of the different avalanche problem types (last paragraph of Section 4.1, third paragraph of Section 5). As highlighted at the end of the discussion section, we regard the dataset to be too small to include more sophisticated functional relationships in our analysis.

We also appreciate the reviewer's comments on alternative or expanded ways for examining our dataset. The reviewer correctly points out that some combinations of avalanche problem types are more likely than others. Prior to this study, our research team actually examined the same dataset of avalanche problem assessments using self-organizing maps to identify common pattern in avalanche problem combinations (Shandro and Haegeli, 2018: https://nhess.copernicus.org/articles/18/1141/2018/). However, since the derivation of these patterns is analytically rather involved and location specific, including it as an additional step in the present analysis makes the results less transparent, more difficult to interpret, and less reproducible. Hence, we believe that our approach of modelling each of the eight avalanche problems independently provides the most insightful and easy to interpret and reproduce contribution.

We agree with the reviewer that regions that are closer together will likely respond to the atmosphere-ocean oscillations more similarly. We included this aspect in our analysis by grouping for forecast areas into larger-scale regions. In our opinion, the current dataset is too small for employing a more sophisticated geospatial modelling approach would require the estimation of additional model parameters, and we are uncertain whether it would offer additional meaningful insight.

**3.6 References biased towards North American studies**
**Referee Comment:**

*The list is impressive. However it is strongly biased towards Canadian studies (or at least studies carried out in North America). This precludes discussing the approach and results in a broad context, and notably to highlight the main strength and weaknesses of what is proposed for a broad readership not especially interested in avalanche regime in western Canada. According to the reference earlier I would suggest the authors to insist more on existing knowledge on avalanche – synoptic patterns relations, avalanche data series and long term changes, and their processing with advanced model based statistical techniques.*

**Author Response:**

Because the impacts of these atmospheric-ocean oscillations are very much location specific, and our description of the effect on western Canada is already quite long, we prefer not to expand our discussion to other geographic regions. However, the make the reader aware that similar studies have been conducted at other locations, we added the following sentence at the end of the second paragraph in the introduction:

*"Similar studies have been conducted in other geographic regions including Iceland (Keylock, 2003) and the Pyrenes in Northern Spain (García-Sellés et al., 2010)."*

While we appreciate the extensive list of references that the reviewer provided, we only included references that explicitly examine the effects of atmosphere-ocean oscillations on avalanche hazard and omitted other papers that focused on synoptic patterns or climate change to prevent the manuscript from becoming even longer.

**3.7 Abstract**

**Referee Comment:**

*Abstract seems way too long for an abstract.*

**Author Response:**

As suggested by the reviewer, we shortened the abstract. Please see the edited manuscript for the updated version of abstract.

**3.8 Organization**

**Referee Comment:**

*Organization is a bit awkward. I assume that sect. 2 content could be easily moved to other sections (introduction, discussion).*

**Author Response:**

While we appreciate this comment, we believe that it is in the realm of personal preferences for writing styles. We chose to have an explicit background section as it allows us to have a more concise introduction that present the reader with the objective of the study before getting into the fine details of some of the concepts. We believe that describing the relevant atmosphere-ocean oscillations affecting the winter weather in western Canada (2.1) and the concept of avalanche problems and its value for describing the nature of avalanche hazard (2.2) are critical pieces of information that the reader needs to understand before reading the method section of our manuscript. Hence, we do not believe that it would be meaningful to move content to the discussion section.

**3.9 Length of paper**

**Referee Comment:**

*The paper is quite lengthy. Given that in my opinion, the main interest lies within the method (avalanche problems as data source and model-based statistical analyses) rather than within the results regarding linkages to synoptic patterns, it could certainly be significantly shortened without losing the key message, and focusing on the main novelties of broad interest.*

**Author Response:**

We appreciate this comment of the reviewer, and in response, we have carefully reviewed the content of the paper. Overall, we believe that the detailed description of our results is necessary for highlighting the validity and value of our approach, and for linking our result to the existing literature in snow hydrology. Please also see our response to comment 3.1 for details on our perspective on the focus of the manuscript.

However, in response to this comment, we deleted the last paragraph of Section 2.2, which described the recent work of Shandro and Haegeli using avalanche problem information to describe the nature and variability of snow and avalanche climates in western Canada. While interesting, this information is not critical for understanding the present analysis.

**3.10 Figures 6-8**

**Referee Comment:**

*Figures 6-8 look a bit too much like direct outputs of a statistical software (namely R probably). I am wondering if something more "visual" and easy to read could be produced? Others figures are nice.*

**Author Response:**

The results of our study are neatly summarized in Figure 4, which present differences in estimated marginal means for each atmosphere-ocean oscillation and region. The intent of Figures 5-8 was to provide the reader with a more detailed perspective on how these differences in marginal means relate to the prevalence observations and the regression models. Since differences in marginal means are an intuitive but fairly processed way to present the results, we believe that including these figures offers the transparency required in an academic publication.

While these figures were created in R, they were custom-built and not the standard output of an existing R package.

Hence, we did not make any changes in response to this comment.

---

## Author Response (AR3)

**Responses to Referee and Editor**

**General comment**

We thank Referee #3 and the editor for reviewing our manuscript again.

**Response to Editor**

We thank Jürg Schweizer for giving our manuscript such a detailed proofread. We have fixed all of the typos that were pointed out.

**Response to Reviewer #3**

We appreciate Referee #3's perspective that our study has a number of contributions that include insight on a geophysical relationship, methodological advances, and operational insight. In our opinion, the focus of our manuscript is on the examination of the relationship between atmosphere-ocean oscillations and the nature of avalanche hazard in western Canada taking advantage of the avalanche problem information published in public avalanche bulletins. We see our statistical modelling approach more as a means to an end and an attempt to use the available data in the best possible way than an explicit methodological contribution. Hence, we feel that the present discussion of our results has the right focus.

While we recognize that TC is an international journal, we feel that a study mainly focusing on the effect of a global phenomenon in a specific location is a meaningful contribution. Our discussion focuses on relating our results to studies examining the effect of the examined oscillations on the winter weather in our study area as we believe this is an important step for highlighting the validity of our approach. In our opinion, explicitly comparing our results to studies in other geographic regions such as Keylock (2003) (Iceland) and Garcia-Selles et al (2010) (Eastern Pyrenees) is challenging due to differences in how avalanche activity is represented and statistical approaches.

Thank you for noticing that we forgot to delete the reference to climate change in the abstract. We fixed the abstract to make it consistent with the content of the manuscript. Not focusing on the potential insight of our results on the effect of climate change on avalanche hazard is also the reason why we did not include references to avalanche research studies that had a climate change connection. In our opinion, Keylock (2003) and Garcia-Selles et al (2010) are the only relevant non-North American studies that exclusively focused on the effects of atmosphere-ocean oscillations.

To further clarify the motivation for examining the link between atmosphere-ocean oscillations and the season nature of avalanche hazard and highlight the operational value, we slightly modified the text in the abstract (addition underlined)

*Since the predictability of the most important atmosphere-ocean oscillations is continuously improving, a better understanding of their effect on avalanche hazard can contribute to the development of informative seasonal avalanche forecasts in a relatively simple way.*

and the conclusion section (addition underlined)

*With the predictability of the most important atmosphere-ocean oscillations continuously improving, this study contributes towards the knowledge necessary for taking advantage of routine atmosphere-ocean oscillation predictions to create informative seasonal avalanche forecasts for western Canada in a relatively simple way.*